# CONTRASTIVE LEARNING WITH HARD NEGATIVE SAMPLES

**Joshua Robinson, Ching-Yao Chuang, Suvrit Sra, Stefanie Jegelka**
Massachusetts Institute of Technology
Cambridge, MA, USA
`{joshrob,cychuang,suvrit,stefje}@mit.edu`

## ABSTRACT

How can you sample good negative examples for contrastive learning? We argue that, as with metric learning, contrastive learning of representations benefits from hard negative samples (i.e., points that are difficult to distinguish from an anchor point). The key challenge toward using hard negatives is that contrastive methods must remain unsupervised, making it infeasible to adopt existing negative sampling strategies that use *true* similarity information. In response, we develop a new family of unsupervised sampling methods for selecting hard negative samples where the user can control the hardness. A limiting case of this sampling results in a representation that tightly clusters each class, and pushes different classes as far apart as possible. The proposed method improves downstream performance across multiple modalities, requires only few additional lines of code to implement, and introduces no computational overhead.

## 1 INTRODUCTION

Owing to their empirical success, contrastive learning methods (Chopra et al., 2005; Hadsell et al., 2006) have become one of the most popular self-supervised approaches for learning representations (Oord et al., 2018; Tian et al., 2019; Chen et al., 2020a). In computer vision, unsupervised contrastive learning methods have even outperformed supervised pre-training for object detection and segmentation tasks (Misra & Maaten, 2020; He et al., 2020).

Contrastive learning relies on two key ingredients: notions of similar (positive) $(x, x^+)$ and dissimilar (negative) $(x, x^-)$ pairs of data points. The training objective, typically *noise-contrastive estimation* (Gutmann & Hyvärinen, 2010), guides the learned representation $f$ to map positive pairs to nearby locations, and negative pairs farther apart; other objectives have also been considered (Chen et al., 2020a). The success of the associated methods depends on the design of informative of the positive and negative pairs, which cannot exploit *true* similarity information since there is no supervision.

Much research effort has addressed sampling strategies for positive pairs, and has been a key driver of recent progress in multi-view and contrastive learning (Blum & Mitchell, 1998; Xu et al., 2013; Bachman et al., 2019; Chen et al., 2020a; Tian et al., 2020). For image data, positive sampling strategies often apply transformations that preserve semantic content, e.g., jittering, random cropping, separating color channels, etc. (Chen et al., 2020a;c; Tian et al., 2019). Such transformations have also been effective in learning control policies from raw pixel data (Srinivas et al., 2020). Positive sampling techniques have also been proposed for sentence, audio, and video data (Logeswaran & Lee, 2018; Oord et al., 2018; Purushwalkam & Gupta, 2020; Sermanet et al., 2018).

Surprisingly, the choice of negative pairs has drawn much less attention in contrastive learning. Often, given an "anchor" point $x$, a "negative" $x^-$ is simply sampled uniformly from the training data, independent of how informative it may be for the learned representation. In supervised and metric learning settings, "hard" (true negative) examples can help guide a learning method to correct its mistakes more quickly (Schroff et al., 2015; Song et al., 2016). For representation learning, informative negative examples are intuitively those pairs that are mapped nearby but should be far apart. This idea is successfully applied in metric learning, where true pairs of dissimilar points are available, as opposed to unsupervised contrastive learning.

---

Code available at: `https://github.com/joshr17/HCL`

With this motivation, we address the challenge of selecting informative negatives for contrastive representation learning. In response, we propose a solution that builds a tunable sampling distribution that prefers negative pairs whose representations are currently very similar. This solution faces two challenges: (1) we do not have access to any true similarity or dissimilarity information; (2) we need an efficient sampling strategy for this tunable distribution. We overcome (1) by building on ideas from positive-unlabeled learning (Elkan & Noto, 2008; Du Plessis et al., 2014), and (2) by designing an efficient, easy to implement importance sampling technique that incurs no computational overhead.

Our theoretical analysis shows that, as a function of the tuning parameter, the optimal representations for our new method place similar inputs in tight clusters, whilst spacing the clusters as far apart as possible. Empirically, our hard negative sampling strategy improves the downstream task performance for image, graph and text data, supporting that indeed, our negative examples are more informative.

**Contributions.** In summary, we make the following contributions:

1. We propose a simple distribution over hard negative pairs for contrastive representation learning, and derive a practical importance sampling strategy with zero computational overhead that takes into account the lack of true dissimilarity information;
2. We theoretically analyze the hard negatives objective and optimal representations, showing that they capture desirable generalization properties;
3. We empirically observe that the proposed sampling method improves the downstream task performance on image, graph and text data.

Before moving onto the problem formulation and our results, we summarize related work below.

## 1.1 RELATED WORK

**Contrastive Representation Learning.** Various frameworks for contrastive learning of visual representations have been proposed, including SimCLR (Chen et al., 2020a;b), which uses augmented views of other items in a minibatch as negative samples, and MoCo (He et al., 2020; Chen et al., 2020c), which uses a momentum updated memory bank of old negative representations to enable the use of very large batches of negative samples. Most contrastive methods are unsupervised, however there exist some that use label information (Sylvain et al., 2020; Khosla et al., 2020). Many works study the role of positive pairs, and, e.g., propose to apply large perturbations for images Chen et al. (2020a;c), or argue to minimize the mutual information within positive pairs, apart from relevant information for the ultimate prediction task (Tian et al., 2020). Beyond visual data, contrastive methods have been developed for sentence embeddings (Logeswaran & Lee, 2018), sequential data (Oord et al., 2018; Hénaff et al., 2020), graph (Sun et al., 2020; Hassani & Khasahmadi, 2020; Li et al., 2019) and node representation learning (Velickovic et al., 2019), and learning representations from raw images for off-policy control (Srinivas et al., 2020). The role of negative pairs hase been much less studied. Chuang et al. (2020) propose a method for "debiasing", i.e., correcting for the fact that not all negative pairs may be true negatives. It does so by taking the viewpoint of Positive-Unlabeled learning, and exploits a decomposition of the true negative distribution. Kalantidis et al. (2020) consider applying Mixup (Zhang et al., 2018) to generate hard negatives in latent space, and Jin et al. (2018) exploit the specific temporal structure of video to generate negatives for object detection.

**Negative Mining in Deep Metric Learning.** As opposed to the contrastive representation learning literature, selection strategies for negative samples have been thoroughly studied in (deep) metric learning (Schroff et al., 2015; Song et al., 2016; Harwood et al., 2017; Wu et al., 2017; Ge, 2018; Suh et al., 2019). Most of these works observe that it is helpful to use negative samples that are difficult for the current embedding to discriminate. Schroff et al. (2015) qualify this, observing that some examples are simply too hard, and propose selecting "semi-hard" negative samples. The well known importance of negative samples in metric learning, where (partial) true dissimilarity information is available, raises the question of negative samples in contrastive learning, the subject of this paper.

## 2 CONTRASTIVE LEARNING SETUP

We begin with the setup and the idea of contrastive representation learning. We wish to learn an embedding $f : \mathcal{X} \to \mathbb{S}^{d-1}/t$ that maps an observation $x$ to a point on a hypersphere $\mathbb{S}^{d-1}/t$ in $\mathbb{R}^d$ of radius $1/t$, where $t$ is the "temperature" scaling hyperparameter.

Following the setup of Arora et al. (2019), we assume an underlying set of discrete latent classes $\mathcal{C}$ that represent semantic content, so that similar pairs $(x, x^+)$ have the same latent class. Denoting

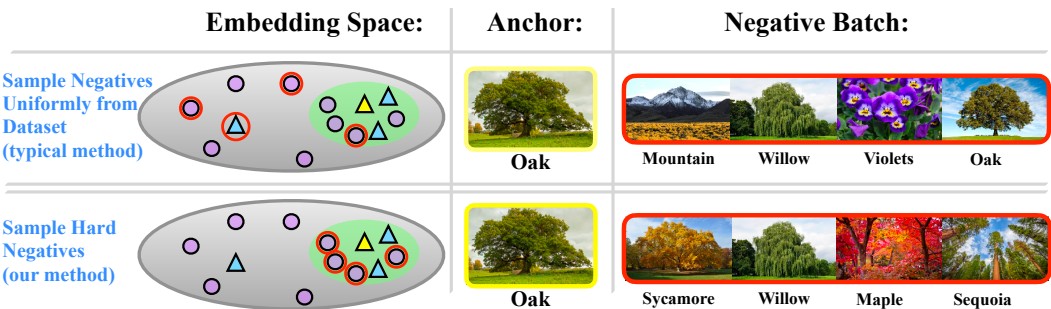

Figure 1: Schematic illustration of negative sampling methods for the example of classifying species of tree. Top row: uniformly samples negative examples (red rings); mostly focuses on very different data points from the anchor (yellow triangle), and may even sample examples from the same class (triangles, vs. circles). Bottom row: Hard negative sampling prefers examples that are (incorrectly) close to the anchor.

the distribution over latent classes by $\rho(c)$ for $c \in \mathcal{C}$, we define the joint distribution $p_{x,c}(x, c) = p(x|c)\rho(c)$ whose marginal $p(x)$ we refer to simply as $p$, and assume $\text{supp}(p) = \mathcal{X}$. For simplicity, we assume $\rho(c) = \tau^+$ is uniform, and let $\tau^- = 1 - \tau^+$ be the probability of another class. Since the class-prior $\tau^+$ is unknown in practice, it must either be treated as a hyperparameter, or estimated (Christoffel et al., 2016; Jain et al., 2016).

Let $h : \mathcal{X} \to \mathcal{C}$ be the true underlying hypothesis that assigns class labels to inputs. We write $x \sim x'$ to denote the label equivalence relation $h(x) = h(x')$. We denote by $p_x^+(x') = p(x'|h(x') = h(x))$, the distribution over points with same label as $x$, and by $p_x^-(x') = p(x'|h(x') \neq h(x))$, the distribution over points with labels different from $x$. We drop the subscript $x$ when the context is clear. Following the usual convention, we overload '$\sim$' and also write $x \sim p$ to denote a point sampled from $p$.

For each data point $x \sim p$, the noise-contrastive estimation (NCE) objective (Gutmann & Hyvärinen, 2010) for learning the representation $f$ uses a *positive* example $x^+$ with the same label as $x$, and *negative* examples $\{x_i^-\}_{i=1}^N$ with (supposedly) different labels, $h(x_i^-) \neq h(x)$, sampled from $q$:

$$\mathbb{E}_{\substack{x \sim p, \ x^+ \sim p_x^+ \\ \{x_i^-\}_{i=1}^N \sim q}} \left[ -\log \frac{e^{f(x)^T f(x^+)}}{e^{f(x)^T f(x^+)} + \frac{Q}{N} \sum_{i=1}^N e^{f(x)^T f(x_i^-)}} \right]. \tag{1}$$

The weighting parameter $Q$ is introduced for the purpose of analysis. When $N$ is finite we take $Q = N$, yielding the usual form of the contrastive objective. The negative sample distribution $q$ is frequently chosen to be the marginal distribution $p$, or, in practice, an empirical approximation of it (Tian et al., 2019; Chen et al., 2020a;c; He et al., 2020; Chen et al., 2020c; Oord et al., 2018; Hénaff et al., 2020). In this paper we ask: is there a better way to choose $q$?

## 3 HARD NEGATIVE SAMPLING

In this section we describe our approach for hard negative sampling. We begin by asking *what makes a good negative sample?* To answer this question we adopt the following two guiding principles:

**Principle 1.** *$q$ should only sample "true negatives" $x_i^-$ whose labels differ from that of the anchor $x$.*

**Principle 2.** *The most useful negative samples are ones that the embedding currently believes to be similar to the anchor.*

In short, negative samples that have different label from the anchor, but that are embedded nearby are likely to be most useful and provide significant gradient information during training. In metric learning there is access to true negative pairs, automatically fulfilling the first principle.

In unsupervised contrastive learning there is no supervision, so upholding Principle 1 is impossible to do exactly. In this paper we propose a method that upholds Principle 1 approximately, and simultaneously combines this idea with the key additional conceptual ingredient of "hardness" (encapsulated in Principle 2). The level of "hardness" in our method can be smoothly adjusted,

allowing the user to select the hardness that best trades-off between an improved learning signal from hard negatives, and the harm due to the correction of false negatives being only approximate. This important since the hardest points are those closest to the anchor, and are expected to have a high propensity to have the same label. Therefore the damage from the approximation not removing all false negatives becomes larger for harder samples, creating the trade-off. As a special case our our method, when the hardness level is tuned fully down, we obtain the method proposed in (Chuang et al., 2020) that only upholds Principle 1 (approximately) but not Principle 2. Finally, beyond Principles 1 and 2, we wish to design an efficient sampling method that does not add additional computational overhead during training.

### 3.1 Proposed Hard Sampling Method

Our first goal is to design a distribution $q$ on $\mathcal{X}$ that is allowed to depend on the embedding $f$ and the anchor $x$. From $q$ we sample a batch of negatives $\{x_i^-\}_{i=1}^N$ according to the principles noted above. We propose sampling negatives from the distribution $q_\beta^-$ defined as

$$q_\beta^-(x^-) := q_\beta(x^-|h(x) \neq h(x^-)), \quad \text{where} \quad q_\beta(x^-) \propto e^{\beta f(x)^\top f(x^-)} \cdot p(x^-),$$

for $\beta \geq 0$. Note that $q_\beta^-$ and $q_\beta$ both depend on $x$, but we suppress the dependance from the notation. The exponential term in $q_\beta$ is an unnormalized von Mises–Fisher distribution with mean direction $f(x)$ and "concentration parameter" $\beta$ (Mardia & Jupp, 2000). There are two key components to $q_\beta^-$, corresponding to each principle: 1) conditioning on the event $\{h(x) \neq h(x^-)\}$ which guarantees that $(x, x^-)$ correspond to different latent classes (Principle 1); 2) the concentration parameter $\beta$ term controls the degree by which $q_\beta$ up-weights points $x^-$ that have large inner product (similarity) to the anchor $x$ (Principle 2). Since $f$ lies on the surface of a hypersphere of radius $1/t$, we have $\|f(x) - f(x')\|^2 = 2/t^2 - 2f(x)^\top f(x')$ so preferring points with large inner product is equivalent to preferring points with small squared Euclidean distance.

Although we have designed $q_\beta^-$ to have all of the desired components, it is not clear how to sample efficiently from it. To work towards a practical method, note that we can rewrite this distribution by adopting a PU-learning viewpoint (Elkan & Noto, 2008; Du Plessis et al., 2014; Chuang et al., 2020). That is, by conditioning on the event $\{h(x) = h(x^-)\}$ we can split $q_\beta(x^-)$ as

$$q_\beta(x^-) = \tau^- q_\beta^-(x^-) + \tau^+ q_\beta^+(x^-), \tag{2}$$

where $q_\beta^+(x^-) = q_\beta(x^-|h(x) = h(x^-)) \propto e^{\beta f(x)^\top f(x^-)} \cdot p^+(x^-)$. Rearranging equation 2 yields a formula $q_\beta^-(x^-) = (q_\beta(x^-) - \tau^+ q_\beta^+(x^-))/\tau^-$ for the negative sampling distribution $q_\beta^-$ in terms of two distributions that are tractable since we have samples from $p$ and can approximate samples from $p^+$ using a set of semantics-preserving transformations, as is typical in contrastive learning methods (see Appendix E for extra discussion of practical implications of this approximation).

It is possible to generate samples from $q_\beta$ and approximately from $q_\beta^+$ using rejection sampling and data augmentations to generate positives. However, rejection sampling involves an algorithmic complication since the procedure for sampling batches must be modified. To avoid this, we instead take an importance sampling approach. To obtain this, first note that fixing the number $Q$ and taking the limit $N \to \infty$ in the objective (1) yields,

$$\mathcal{L}(f, q) = \mathbb{E}_{\substack{x \sim p \\ x^+ \sim p_x^+}} \left[ -\log \frac{e^{f(x)^T f(x^+)}}{e^{f(x)^T f(x^+)} + Q\mathbb{E}_{x^- \sim q}[e^{f(x)^T f(x^-)}]} \right]. \tag{3}$$

The original objective (1) can be viewed as a finite negative sample approximation to $\mathcal{L}(f, q)$ (note implicitly $\mathcal{L}(f, q)$ depends on $Q$) . Inserting $q = q_\beta^-$ and using the rearrangement of equation (2) we obtain the following hardness-biased objective:

$$\mathbb{E}_{\substack{x \sim p \\ x^+ \sim p_x^+}} \left[ -\log \frac{e^{f(x)^T f(x^+)}}{e^{f(x)^T f(x^+)} + \frac{Q}{\tau^-}(\mathbb{E}_{x^- \sim q_\beta}[e^{f(x)^T f(x^-)}] - \tau^+ \mathbb{E}_{v \sim q_\beta^+}[e^{f(x)^T f(v)}])} \right]. \tag{4}$$

This objective suggests that we need only to approximate *expectations* $\mathbb{E}_{x^- \sim q_\beta}[e^{f(x)^T f(x^-)}]$ and $\mathbb{E}_{v \sim q_\beta^+}[e^{f(x)^T f(v)}]$ over $q_\beta$ and $q_\beta^+$ (rather than explicitly sampling). This can be achieved using

classical Monte-Carlo importance sampling techniques using samples from $p$ and $p^+$ as follows:

$$\mathbb{E}_{x^- \sim q_\beta}[e^{f(x)^T f(x^-)}] = \mathbb{E}_{x^- \sim p}[e^{f(x)^T f(x^-)} q_\beta / p] = \mathbb{E}_{x^- \sim p}[e^{(\beta+1)f(x)^T f(x^-)} / Z_\beta],$$

$$\mathbb{E}_{v \sim q_\beta^+}[e^{f(x)^T f(v)}] = \mathbb{E}_{v \sim p^+}[e^{f(x)^T f(v)} q_\beta^+ / p^+] = \mathbb{E}_{v \sim p^+}[e^{(\beta+1)f(x)^T f(v)} / Z_\beta^+],$$

where $Z_\beta, Z_\beta^+$ are the partition functions of $q_\beta$ and $q_\beta^+$ respectively. The right hand terms readily admit empirical approximations by replacing $p$ and $p^+$ with $\hat{p}(x) = \frac{1}{N} \sum_{i=1}^{N} \delta_{x_i^-}(x)$ and $\hat{p}^+(x) = \frac{1}{M} \sum_{i=1}^{M} \delta_{x_i^+}(x)$ respectively ($\delta_w$ denotes the Dirac delta function centered at $w$). The only unknowns left are the partition functions, $Z_\beta = \mathbb{E}_{x^- \sim p}[e^{\beta f(x)^T f(x^-)}]$ and $Z_\beta^+ = \mathbb{E}_{x^+ \sim p^+}[e^{\beta f(x)^T f(x^+)}]$ which themselves are expectations over $p$ and $p^+$ and therefore admit empirical estimates,

$$\widehat{Z}_\beta = \frac{1}{N} \sum_{i=1}^{N} e^{\beta f(x)^\top f(x_i^-)}, \qquad \widehat{Z}_\beta^+ = \frac{1}{M} \sum_{i=1}^{M} e^{\beta f(x)^\top f(x_i^+)}.$$

It is important to emphasize the simplicity of the implementation of our proposed approach. Since we propose to reweight the objective instead of modifying the sampling procedure, only two extra lines of code are needed to implement our approach, with no additional computational overhead. PyTorch-style pseudocode for the objective is given in Fig. 13 in Appendix D.

## 4 ANALYSIS OF HARD NEGATIVE SAMPLING

### 4.1 HARD SAMPLING INTERPOLATES BETWEEN MARGINAL AND WORST-CASE NEGATIVES

Intuitively, the concentration parameter $\beta$ in our proposed negative sample distribution $q_\beta^-$ controls the level of "hardness" of the negative samples. As discussed earlier, the debiasing method of Chuang et al. (2020) can be recovered as a special case: taking $\beta = 0$ to obtain the distribution $q_0^-$. This case amounts to correcting for the fact that some samples in a negative batch sampled from $p$ will have the same label as the anchor. But what interpretation does large $\beta$ admit? Specifically, what does the distribution $q_\beta^-$ converge to in the limit $\beta \to \infty$, if anything? We show that in the limit $q_\beta^-$ approximates an inner solution to the following zero-sum two player game.

$$\inf_f \sup_{q \in \Pi} \left\{ \mathcal{L}(f, q) = \mathbb{E}_{\substack{x \sim p \\ x^+ \sim p_x^+}} \left[ -\log \frac{e^{f(x)^T f(x^+)}}{e^{f(x)^T f(x^+)} + Q \mathbb{E}_{x^- \sim q}[e^{f(x)^T f(x^-)}]} \right] \right\}. \tag{5}$$

where $\Pi = \{ q = q(\cdot; x, f) : \text{supp}(q(\cdot; x, f)) \subseteq \{x' \in \mathcal{X} : x' \nsim x\}, \forall x \in \mathcal{X} \}$ is the set of distributions with support that is disjoint from points with the same class as $x$ (without loss of generality we assume $\{x' \in \mathcal{X} : x' \nsim x\}$ is non-empty). Since $q = q(\cdot; x, f)$ depends on $x$ and $f$ it can be thought of as a family of distributions. The formal statement is as follows.

**Proposition 3.** *Let $\mathcal{L}^*(f) = \sup_{q \in \Pi} \mathcal{L}(f, q)$. Then for any $t > 0$ and $f : \mathcal{X} \to \mathbb{S}^{d-1}/t$ we observe the convergence $\mathcal{L}(f, q_\beta^-) \longrightarrow \mathcal{L}^*(f)$ as $\beta \to \infty$.*

*Proof.* See Appendix A.1. □

To develop a better intuitive understanding of the worst case negative distribution objective $\mathcal{L}^*(f) = \sup_{q \in \Pi} \mathcal{L}(f, q)$, we note that the supremum can be characterized analytically. Indeed,

$$\sup_{q \in \Pi} \mathcal{L}(f, q) = -\mathbb{E}_{\substack{x \sim p \\ x^+ \sim p_x^+}} f(x)^T f(x^+) + \sup_{q \in \Pi} \mathbb{E}_{\substack{x \sim p \\ x^+ \sim p_x^+}} \log \left\{ e^{f(x)^T f(x^+)} + Q \mathbb{E}_{x^- \sim q}[e^{f(x)^T f(x^-)}] \right\}$$

$$= -\mathbb{E}_{\substack{x \sim p \\ x^+ \sim p_x^+}} f(x)^T f(x^+) + \mathbb{E}_{\substack{x \sim p \\ x^+ \sim p_x^+}} \log \left\{ e^{f(x)^T f(x^+)} + Q \cdot \sup_{q \in \Pi} \mathbb{E}_{x^- \sim q}[e^{f(x)^T f(x^-)}] \right\}.$$

The supremum over $q$ can be pushed inside the expectation since $q$ is a family of distribution indexed by $x$, reducing the problem to maximizing $\mathbb{E}_{x^- \sim q}[e^{f(x)^T f(x^-)}]$, which is solved by any $q^*$ whose support is a subset of $\arg \sup_{x^- : x^- \nsim x} e^{f(x)^T f(x^-)}$ if the supremum is attained. However, computing such points involves maximizing a neural network. Instead of taking this challenging route, using $q_\beta^-$ defines a lower bound by placing higher probability on $x^-$ for which $f(x)^T f(x^-)$ is large. This lower bound becomes tight as $\beta \to \infty$ (Proposition 3).

## 4.2 Optimal Embeddings on the Hypersphere for Worst-Case Negative Samples

What desirable properties does an optimal contrastive embedding (global minimizer of $\mathcal{L}$) possess that make the representation generalizable? To study this question, we first analyze the distribution of an optimal embedding $f^*$ on the hypersphere when negatives are sampled from the adversarial worst-case distribution. We consider a different limiting viewpoint of objective (1) as the number of negative samples $N \to \infty$. Following the formulation of Wang & Isola (2020) we take $Q = N$ in (1), and subtract $\log N$. This changes neither the set of minimizers, nor the geometry of the loss surface. Taking the number of negative samples $N \to \infty$ yields the limiting objective,

$$\mathcal{L}_\infty(f, q) = \mathbb{E}_{\substack{x \sim p \\ x^+ \sim p_x^+}} \left[ -\log \frac{e^{f(x)^T f(x^+)}}{\mathbb{E}_{x^- \sim q}[e^{f(x)^T f(x^-)}]} \right]. \tag{6}$$

**Theorem 4.** *Suppose the downstream task is classification (i.e. $\mathcal{C}$ is finite), and let $\mathcal{L}_\infty^*(f) = \sup_{q \in \Pi} \mathcal{L}_\infty(f, q)$. The infimum $\inf_{f: \text{ measurable}} \mathcal{L}_\infty^*(f)$ is attained, and any $f^*$ achieving the global minimum is such that $f^*(x) = f^*(x^+)$ almost surely. Furthermore, letting $\mathbf{v}_c = f^*(x)$ for any $x$ such that $h(x) = c$ (so $\mathbf{v}_c$ is well defined up to a set of $x$ of measure zero), $f^*$ is characterized as being any solution to the following ball-packing problem,*

$$\max_{\{\mathbf{v}_c \in \mathbb{S}^{d-1}/t\}_{c \in \mathcal{C}}} \sum_{c \in \mathcal{C}} \rho(c) \cdot \min_{c' \neq c} \|\mathbf{v}_c - \mathbf{v}_{c'}\|^2. \tag{7}$$

*Proof.* See Appendix A.2. □

**Interpretation:** The first component of the result is that $f^*(x) = f^*(x^+)$ almost surely for an optimal $f^*$. That is, an optimal embedding $f^*$ must be invariant across pairs of similar inputs $x, x^+$. The second component is characterizing solutions via the classical geometrical Ball-Packing Problem of Tammes (1930) (Eq. 7) that has only been solved exactly for uniform $\rho$, for specific of $|\mathcal{C}|$ and typically for $\mathbb{S}^2$ (Schütte & Van der Waerden, 1951; Musin & Tarasov, 2015; Tammes, 1930). When the distribution $\rho$ over classes is uniform this problem is solved by a set of $|\mathcal{C}|$ points on the hypersphere such that the average squared-$\ell_2$ distance from a point to the nearest other point is as large as possible. In other words, suppose we wish to place $|\mathcal{C}|$ number of balls[1] on $\mathbb{S}^{d-1}$ so that they do not intersect. Then solutions to Tammes' Problem (7) expresses (twice) the largest possible average squared radius that the balls can have. So, we have a ball-packing problem where instead of trying to pack as many balls as possible of a fixed size, we aim to pack a fixed number of balls (one for each class) to have as big radii as possible. Non-uniform $\rho$ adds importance weights to each fixed ball. In summary, solutions of the problem $\min_f \mathcal{L}_\infty^*(f)$ are a maximum margin clustering.

This understanding of global minimizers of $\mathcal{L}_\infty^*(f) = \sup_{q \in \Pi} \mathcal{L}_\infty(f, q)$ can further developed into a better understanding of generalization on downstream tasks. The next result shows that representations that achieve small excess risk on the objective $\mathcal{L}_\infty^*$ still separate clusters well in the sense that a simple 1-nearest neighbor classifier achieves low classification error.

**Theorem 5.** *Suppose $\rho$ is uniform on $\mathcal{C}$ and $f$ is such that $\mathcal{L}_\infty^*(f) - \inf_{\bar{f} \text{ measurable}} \mathcal{L}_\infty^*(\bar{f}) \leq \varepsilon$ with $\varepsilon \leq 1$. Let $\{\mathbf{v}_c^* \in \mathbb{S}^{d-1}/t\}_{c \in \mathcal{C}}$ be a solution to Problem 7, and define the constant $\xi = \min_{c, c^- : c \neq c^-} \|\mathbf{v}_c^* - \mathbf{v}_{c-}^*\| > 0$. Then there exists a set of vectors $\{\mathbf{v}_c \in \mathbb{S}^{d-1}/t\}_{c \in \mathcal{C}}$ such that the 1-nearest neighbor classifier $\hat{h}(x) = \arg\min_{\bar{c} \in \mathcal{C}} \|f(x) - \mathbf{v}_{\bar{c}}\|$ (ties broken arbitrarily) achieves misclassification risk,*

$$\mathbb{P}_{x,c}(\hat{h}(x) \neq c) \leq \frac{8\varepsilon}{(\xi^2 - 2|\mathcal{C}|(1 + 1/t)\varepsilon^{1/2})^2}$$

*Proof.* See Appendix A.3. □

In particular, $\mathbb{P}(\hat{h}(x) \neq c) = \mathcal{O}(\varepsilon)$ as $\varepsilon \to 0$, and in the limit $\varepsilon \to 0$ we recover the invariance claim of Theorem 4 as a special case. The result can be generalized to arbitrary $\rho$ by replacing $|\mathcal{C}|$ in the bound by $1/\min_c \rho(c)$. The result also implies that it is possible to build simple classifiers for tasks that involve only a subset of classes from $\mathcal{C}$, or classes that are a union of classes from $\mathcal{C}$. The constant $\xi = \min_{c, c^- : c \neq c^-} \|\mathbf{v}_c^* - \mathbf{v}_{c-}^*\| > 0$ is a purely geometrical property of spheres, and describes the minimum separation distance between a set of points that solves the Tammes' ball-packing problem.

---

[1]For a manifold $\mathcal{M} \subseteq \mathbb{R}^d$, we say $C \subset \mathcal{M}$ is a ball if it is connected, and there exists a Euclidean ball $\mathcal{B} = \{x \in \mathbb{R}^d : \|x\|_2 \leq R\}$ for which $C = \mathcal{M} \cap \mathcal{B}$.

## 5 EMPIRICAL RESULTS

Next, we evaluate our hard negative sampling method empirically, and apply it as a modification to state-of-the-art contrastive methods on image, graph, and text data. For all experiments $\beta$ is treated as a hyper-parameter (see ablations in Fig. 2 for more understanding of how to pick $\beta$). Values for $M$ and $\tau^+$ must also be determined. We fix $M = 1$ for all experiments, since taking $M > 1$ would increase the number of inputs for the forward-backward pass. Lemma 11 in the appendix gives a theoretical justification for the choice of $M = 1$. Choosing the class-prior $\tau^+$ can be done in two ways: estimating it from data (Christoffel et al., 2016; Jain et al., 2016), or treating it as a hyper-parameter. The first option requires the possession of labeled data *before* contrastive training.

### 5.1 IMAGE REPRESENTATIONS

We begin by testing the hard sampling method on vision tasks using the STL10, CIFAR100 and CIFAR10 data. We use SimCLR (Chen et al., 2020a) as the baseline method, and all models are trained for 400 epochs. The results in Fig. 2 show consistent improvement over SimCLR ($q = p$) and the particular case of our method with $\beta = 0$ proposed in (Chuang et al., 2020) (called debiasing) on STL10 and CIFAR100. For $N = 510$ negative examples per data point we observe absolute improvements of 3% and 7.3% over SimCLR on CIFAR100 and STL10 respectively, and absolute improvements over the best debiased baseline of 1.9% and 3.2%. On tinyImageNet (Tab. 1) we observe an absolute improvement of 3.6% over SimCLR, while on CIFAR10 there is a slight improvement for smaller $N$, which disappears at larger $N$. See Appendix C.1 results using MoCo-v2 for large negative batch size, and Appendix D.1 for full setup details.

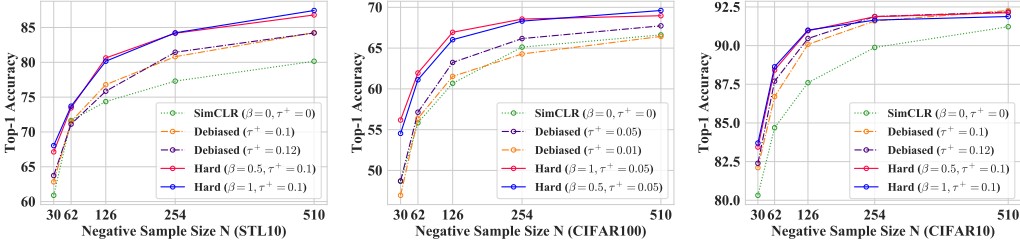

Figure 2: **Classification accuracy on downstream tasks.** Embeddings trained using hard, debiased, and standard ($\beta = 0, \tau^+ = 0$) versions of SimCLR, and evaluated using linear readout accuracy.

### 5.2 GRAPH REPRESENTATIONS

Second, we consider hard negative sampling in the context of learning graph representations. We use the state-of-the-art InfoGraph method introduced by Sun et al. (2020) as the baseline, which is suitable for downstream graph-level classification. The objective is of a slightly different form from the NCE loss. Because of this we use a generalization of the formulation presented in Section 3 (See Appendix B for details). In doing so, we illustrate that it is easy to adapt our hard sampling method to other contrastive frameworks.

| SimCLR | Debiased | Hard ($\beta = 1$) |
| --- | --- | --- |
| 53.4% | 53.7% | 57.0% |

Table 1: Top-1 linear readout on tinyImageNet. Class prior is set to $\tau^+ = 0.01$.

Fig. 3 shows the results of fine-tuning an SVM (Boser et al., 1992; Cortes & Vapnik, 1995) on the fixed, learned embedding for a range of different values of $\beta$. Hard sampling does as well as InfoGraph in all cases, and better in 6 out of 8 cases. For ENZYMES and REDDIT, hard negative samples improve the accuracy by 3.2% and 2.4%, respectively, for DD and PTC by $1 - 2\%$, and for IMDB-B and MUTAG by at least 0.5%. Usually, multiple different choices of $\beta > 0$ were competitive with the InfoGraph baseline: 17 out of the 24 values of $\beta > 0$ tried (across all 8 datasets) achieve accuracy as high or better than InfoGraph ($\beta = 0$).

### 5.3 SENTENCE REPRESENTATIONS

Third, we test hard negative sampling on learning representations of sentences using the *quick-thoughts* (QT) vectors framework introduced by Logeswaran & Lee (2018), which uses adjacent sentences (before/after) as positive samples. Embeddings are trained using the unlabeled BookCorpus dataset (Kiros et al., 2015), and evaluated following the protocol of Logeswaran & Lee (2018) on six downstream tasks. The results are reported in Table 2. Hard sampling outperforms or equals the QT

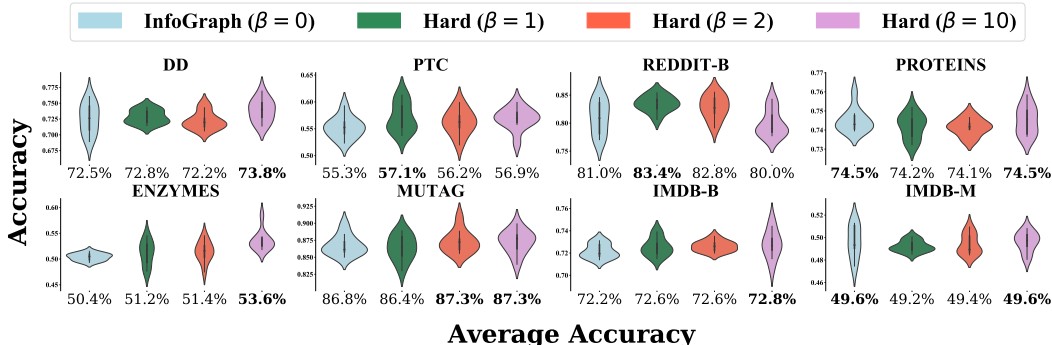

Figure 3: **Classification accuracy on downstream tasks.** We compare graph representations on four classification tasks. Accuracies are obtained by fine-tuning an SVM readout function, and are the average of 10 runs, each using 10-fold cross validation. Results in **bold** indicate best performer.

baseline in 5 out of 6 cases, the debiased baseline (Chuang et al., 2020) in 4 out of 6, and both in 3 out of 6 cases. Setting $\tau^+ > 0$ led to numerical issues in optimization for hard sampling.

| Objective | MR | CR | SUBJ | MPQA | TREC | MSRP (Acc) | (F1) |
|---|---|---|---|---|---|---|---|
| QT ($\beta = 0, \tau^+ = 0$) | 76.8 | 81.3 | 86.6 | 93.4 | **89.8** | 73.6 | 81.8 |
| Debiased ($\tau^+ = 0.01$) | 76.2 | 82.9 | 86.9 | **93.7** | 89.1 | **74.7** | **82.7** |
| Hard ($\beta = 1, \tau^+ = 0$) | 77.1 | 82.5 | **87.0** | 92.9 | 89.2 | 73.9 | 82.2 |
| Hard ($\beta = 2, \tau^+ = 0$) | **77.4** | **83.6** | 86.8 | 93.4 | 88.7 | 73.5 | 82.0 |

Table 2: **Classification accuracy on downstream tasks.** Sentence representations are learned using quick-thoughts (QT) vectors on the BookCorpus dataset and evaluated on six classification tasks. Evaluation of binary classification tasks (MR, CR, SUBJ, MPQA) uses 10-fold cross validation.

## 6 A CLOSER LOOK AT HARD SAMPLING

### 6.1 ARE HARDER SAMPLES NECESSARILY BETTER?

By setting $\beta$ to large values, one can focus on only the hardest samples in a training batch. But is this desirable? Fig. 4 (left, middle) shows that for vision problems, taking larger $\beta$ does not necessarily lead to better representations. In contrast, when one uses true positive pairs during training (green curve, uses label information for positive but not negative pairs), the downstream performance monotonically increases with $\beta$ until convergence (Fig. 4, middle). Interestingly, this is achieved without using label information for the negative pairs. This observation suggests an explanation for why bigger $\beta$ hurts performance in practice. Debiasing (conditioning on the event $\{h(x) \neq h(x^-)\}$) using the true $p^+$ corrects for sampling $x^-$ with the same label as $x$. However, since in practice we approximate $p^+$ using a set of data transformations, we can only partially correct. This is harmful for large $\beta$ since this regime strongly prefers $x^-$ for which $f(x^-)$ is close to $f(x)$, many of whom will have the same label as $x$ if not corrected for. We note also that by annealing $\beta$ (gradually decreasing $\beta$ to 0 throughout training; see Appendix D.1 for details) it is possible to be more robust to the choice of initial $\beta$, with marginal impact on downstream accuracy compared to the best fixed value of $\beta$.

### 6.2 DOES AVOIDING FALSE NEGATIVES IMPROVE HARD SAMPLING?

Our proposed hard negative sampling method conditions on the event $\{h(x) \neq h(x^-)\}$ in order to avoid false negatives (termed "debiasing" (Chuang et al., 2020)). But does this help? To test this, we train four embeddings: hard sampling with and without debiasing, and uniform sampling ($\beta = 0$) with and without debiasing. The results in Fig. 4 (right) show that hard sampling with debiasing obtains the highest linear readout accuracy on STL10, only using hard sampling or only debiasing yields (in this case) similar accuracy. All improve over the SimCLR baseline.

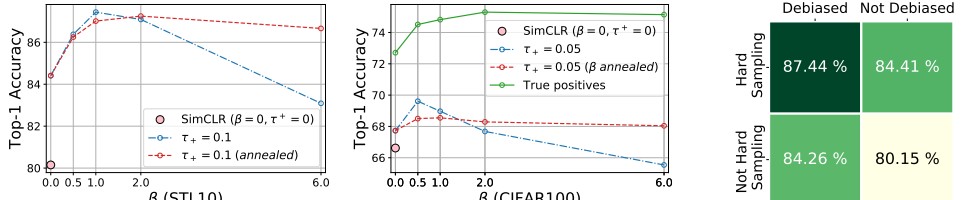

Figure 4: Left: the effect of varying concentration parameter $\beta$ on linear readout accuracy. Middle: linear readout accuracy as concentration parameter $\beta$ varyies, in the case of contrastive learning (fully unsupervised), using true positive samples (uses label information), and an annealing method that improves robustness to the choice of $\beta$ (see Appendix D.1 for details). Right: STL10 linear readout accuracy for hard sampling with and without debiasing, and non-hard sampling ($\beta = 0$) with and without debiasing. Best results come from using both simultaneously.

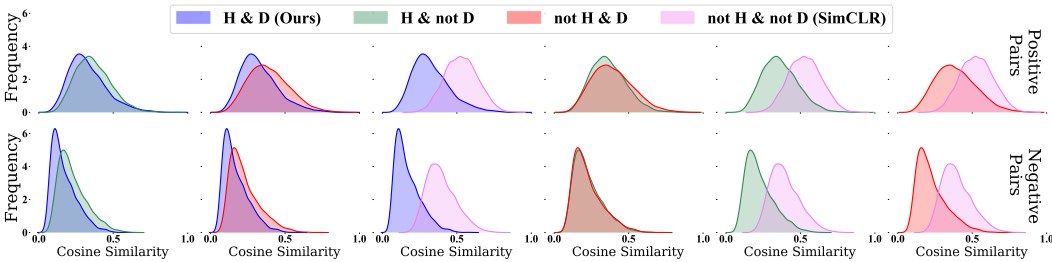

Figure 5: Histograms of cosine similarity of pairs of points with the same label (top) and different labels (bottom) for embeddings trained on STL10 with four different objectives. H=Hard Sampling, D=Debiasing. Histograms overlaid pairwise to allow for convenient comparison.

Fig. 5 compares the histograms of cosine similarities of positive and negative pairs for the four learned representations. The representation trained with hard negatives and debiasing assigns much lower similarity score to a pair of negative samples than other methods. On the other hand, the SimCLR baseline assigns higher cosine similarity scores to pairs of positive samples. However, to discriminate positive and negative pairs, a key property is the amount of *overlap* of positive and negative histograms. Our hard sampling method achieves less overlap than SimCLR, by better trading off higher dissimilarity of negative pairs with less similarity of positive pairs. Similar tradeoffs are observed for the debiased objective, and hard sampling without debiasing.

## 6.3 HOW DO HARD NEGATIVES AFFECT OPTIMIZATION?

Fig. 11 (in Appendix C due to space constraints) shows the performance on STL10 and CIFAR100 of SimCLR versus using hard negatives throughout training. We use weighted $k$-nearest neighbors with $k = 200$ as the classifier and evaluate each model once every five epochs. Hard sampling with $\beta = 1$ leads to much faster training: on STL10 hard sampling takes only 60 epochs to reach the same performance as SimCLR does in 400 epochs. On CIFAR100 hard sampling takes only 125 epochs to reach the same performance as SimCLR does in 400 epochs. We speculate that the speedup is, in part, due to hard negatives providing non-negligible gradient information during training.

## 7 CONCLUSION

We argue for the value of hard negatives in unsupervised contrastive representation learning, and introduce a simple hard negative sampling method. Our work connects two major lines of work: contrastive learning, and negative mining in metric learning. Doing so requires overcoming an apparent roadblock: negative mining in metric learning uses pairwise similarity information as a core component, while contrastive learning is unsupervised. Our method enjoys several nice aspects: having desirable theoretical properties, a very simple implementation that requires modifying only a couple of lines of code, not changing anything about the data sampling pipeline, introducing zero extra computational overhead, and handling false negatives in a principled way.

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

## A  Analysis of Hard Sampling

### A.1  Hard Sampling Interpolates Between Marginal and Worst-Case Negatives

We begin by proving Proposition 3. Recall that the proposition stated the following.

**Proposition 6.** *Let $\mathcal{L}^*(f) = \sup_{q \in \Pi} \mathcal{L}(f, q)$. Then for any $t > 0$ and measurable $f : \mathcal{X} \to \mathbb{S}^{d-1}/t$ we observe the convergence $\mathcal{L}(f, q_\beta^-) \longrightarrow \mathcal{L}^*(f)$ as $\beta \to \infty$.*

*Proof.* Consider the following essential supremum,

$$M(x) = \operatorname*{ess\,sup}_{x^- \in \mathcal{X}: x^- \nsim x} f(x)^T f(x^-) = \sup\{m > 0 : m \geq f(x)^T f(x^-) \text{ a.s. for } x^- \sim p^-\}.$$

The second inequality holds since $\mathrm{supp}(p) = \mathcal{X}$. We may rewrite

$$\mathcal{L}^*(f) = \mathbb{E}_{\substack{x \sim p \\ x^+ \sim p_x^+}} \left[ -\log \frac{e^{f(x)^T f(x^+)}}{e^{f(x)^T f(x^+)} + Q e^{M(x)}} \right],$$

$$\mathcal{L}(f, q_\beta^-) = \mathbb{E}_{\substack{x \sim p \\ x^+ \sim p_x^+}} \left[ -\log \frac{e^{f(x)^T f(x^+)}}{e^{f(x)^T f(x^+)} + Q \mathbb{E}_{x^- \sim q_\beta^-}[e^{f(x)^T f(x^-)}]} \right].$$

The difference between these two terms can be bounded as follows,

$$\left| \mathcal{L}^*(f) - \mathcal{L}(f, q_\beta^-) \right| \leq \mathbb{E}_{\substack{x \sim p \\ x^+ \sim p_x^+}} \left| -\log \frac{e^{f(x)^T f(x^+)}}{e^{f(x)^T f(x^+)} + Q e^{M(x)}} + \log \frac{e^{f(x)^T f(x^+)}}{e^{f(x)^T f(x^+)} + Q \mathbb{E}_{x^- \sim q_\beta^-}[e^{f(x)^T f(x^-)}]} \right|$$

$$= \mathbb{E}_{\substack{x \sim p \\ x^+ \sim p_x^+}} \left| \log \left( e^{f(x)^T f(x^+)} + Q \mathbb{E}_{x^- \sim q_\beta^-}[e^{f(x)^T f(x^-)}] \right) - \log \left( e^{f(x)^T f(x^+)} + Q e^{M(x)} \right) \right|$$

$$\leq \frac{e^{1/t}}{Q+1} \cdot \mathbb{E}_{\substack{x \sim p \\ x^+ \sim p_x^+}} \left| e^{f(x)^T f(x^+)} + Q \mathbb{E}_{x^- \sim q_\beta^-}[e^{f(x)^T f(x^-)}] - e^{f(x)^T f(x^+)} - Q e^{M(x)} \right|$$

$$= \frac{e^{1/t} Q}{Q+1} \cdot \mathbb{E}_{x \sim p} \left| \mathbb{E}_{x^- \sim q_\beta^-}[e^{f(x)^T f(x^-)}] - e^{M(x)} \right|$$

$$\leq e^{1/t} \cdot \mathbb{E}_{x \sim p} \mathbb{E}_{x^- \sim q_\beta^-} \left| e^{M(x)} - e^{f(x)^T f(x^-)} \right|$$

where for the second inequality we have used the fact that $f$ lies on the hypersphere of radius $1/t$ to restrict the domain of the logarithm to values greater than $(Q+1)e^{-1/t}$. Because of this the logarithm is Lipschitz with parameter $e^{1/t}/(Q+1)$. Using again the fact that $f$ lies on the hypersphere we know that $|f(x)^T f(x^-)| \leq 1/t^2$ and hence have the following inequality,

$$\mathbb{E}_{x \sim p} \mathbb{E}_{q_\beta^-} \left| e^{M(x)} - e^{f(x)^T f(x^-)} \right| \leq e^{1/t^2} \mathbb{E}_{x \sim p} \mathbb{E}_{q_\beta^-} \left| M(x) - f(x)^T f(x^-) \right|$$

Let us consider the inner expectation $E_\beta(x) = \mathbb{E}_{q_\beta^-} |M(x) - f(x)^T f(x^-)|$. Note that since $f$ is bounded, $E_\beta(x)$ is uniformly bounded in $x$. Therefore, in order to show the convergence $\mathcal{L}(f, q_\beta^-) \to \mathcal{L}^*(f)$ as $\beta \to \infty$, it suffices by the dominated convergence theorem to show that $E_\beta(x) \to 0$ pointwise as $\beta \to \infty$ for arbitrary fixed $x \in \mathcal{X}$.

From now on we denote $M = M(x)$ for brevity, and consider a fixed $x \in \mathcal{X}$. From the definition of $q_\beta^-$ it is clear that $q_\beta^- \ll p^-$. That is, since $q_\beta^- = c \cdot p^-$ for some (non-constant) $c$, it is absolutely continuous with respect to $p^-$. So $M(x) \geq f(x)^T f(x^-)$ almost surely for $x^- \sim q_\beta^-$, and we

may therefore drop the absolute value signs from our expectation. Define the following event $\mathcal{G}_\varepsilon = \{x^- : f(x)^\top f(x^-) \geq M - \varepsilon\}$ where $\mathcal{G}$ is refers to a "good" event. Define its complement $\mathcal{B}_\varepsilon = \mathcal{G}_\varepsilon^c$ where $\mathcal{B}$ is for "bad". For a fixed $x \in \mathcal{X}$ and $\varepsilon > 0$ consider,

$$
\begin{aligned}
E_\beta(x) &= \mathbb{E}_{x^- \sim q_\beta^-} \left| M(x) - f(x)^T f(x^-) \right| \\
&= \mathbb{P}_{x^- \sim q_\beta^-}(\mathcal{G}_\varepsilon) \cdot \mathbb{E}_{x^- \sim q_\beta^-} \left[ \left| M(x) - f(x)^T f(x^-) \right| | \mathcal{G}_\varepsilon \right] + \mathbb{P}_{x^- \sim q_\beta^-}(\mathcal{B}_\varepsilon) \cdot \mathbb{E}_{x^- \sim q_\beta^-} \left[ \left| M(x) - f(x)^T f(x^-) \right| | \mathcal{B}_\varepsilon \right] \\
&\leq \mathbb{P}_{x^- \sim q_\beta^-}(\mathcal{G}_\varepsilon) \cdot \varepsilon + 2\mathbb{P}_{x^- \sim q_\beta^-}(\mathcal{B}_\varepsilon) \\
&\leq \varepsilon + 2\mathbb{P}_{x^- \sim q_\beta^-}(\mathcal{B}_\varepsilon).
\end{aligned}
$$

We need to control $\mathbb{P}_{x^- \sim q_\beta^-}(\mathcal{B}_\varepsilon)$. Expanding,

$$
\mathbb{P}_{x^- \sim q_\beta^-}(\mathcal{B}_\varepsilon) = \int_{\mathcal{X}} \mathbf{1}\left\{ f(x)^T f(x^-) < M(x) - \varepsilon \right\} \frac{e^{\beta f(x)^T f(x^-)} \cdot p^-(x^-)}{Z_\beta} \mathrm{d}x^-
$$

where $Z_\beta = \int_{\mathcal{X}} e^{\beta f(x)^T f(x^-)} p^-(x^-) \mathrm{d}x^-$ is the partition function of $q_\beta^-$. We may bound this expression by,

$$
\begin{aligned}
\int_{\mathcal{X}} \mathbf{1}\left\{ f(x)^T f(x^-) < M - \varepsilon \right\} \frac{e^{\beta(M-\varepsilon)} \cdot p^-(x^-)}{Z_\beta} \mathrm{d}x^- &\leq \frac{e^{\beta(M-\varepsilon)}}{Z_\beta} \int_{\mathcal{X}} \mathbf{1}\left\{ f(x)^T f(x^-) < M - \varepsilon \right\} p^-(x^-) \mathrm{d}x^- \\
&= \frac{e^{\beta(M-\varepsilon)}}{Z_\beta} \mathbb{P}_{x^- \sim p^-}(\mathcal{B}_\varepsilon) \\
&\leq \frac{e^{\beta(M-\varepsilon)}}{Z_\beta}
\end{aligned}
$$

Note that

$$
Z_\beta = \int_{\mathcal{X}} e^{\beta f(x)^T f(x^-)} p^-(x^-) \mathrm{d}x^- \geq e^{\beta(M-\varepsilon/2)} \mathbb{P}_{x^- \sim p^-}(f(x)^T f(x^-) \geq M - \varepsilon/2).
$$

By the definition of $M = M(x)$ the probability $\rho_\varepsilon = \mathbb{P}_{x^- \sim p^-}(f(x)^T f(x^-) \geq M - \varepsilon/2) > 0$, and we may therefore bound,

$$
\begin{aligned}
\mathbb{P}_{x^- \sim q_\beta^-}(\mathcal{B}_\varepsilon) &= \frac{e^{\beta(M-\varepsilon)}}{e^{\beta(M-\varepsilon/2)} \rho_\varepsilon} \\
&= e^{-\beta\varepsilon/2}/\rho_\varepsilon \\
&\longrightarrow 0 \quad \text{as} \ \beta \to \infty.
\end{aligned}
$$

We may therefore take $\beta$ to be sufficiently big so as to make $\mathbb{P}_{x^- \sim q_\beta^-}(\mathcal{B}_\varepsilon) \leq \varepsilon$ and therefore $E_\beta(x) \leq 3\varepsilon$. In other words, $E_\beta(x) \longrightarrow 0$ as $\beta \to \infty$. $\qquad\square$

## A.2 OPTIMAL EMBEDDINGS ON THE HYPERSPHERE FOR WORST-CASE NEGATIVE SAMPLES

In order to study properties of global optima of the contrastive objective using the adversarial worst case hard sampling distribution recall that we have the following limiting objective,

$$
\mathcal{L}_\infty(f, q) = \mathbb{E}_{\substack{x \sim p \\ x^+ \sim p_x^+}} \left[ -\log \frac{e^{f(x)^T f(x^+)}}{\mathbb{E}_{x^- \sim q_\beta} [e^{f(x)^T f(x^-)}]} \right]. \tag{8}
$$

We may separate the logarithm of a quotient into the sum of two terms plus a constant,

$$
\mathcal{L}_\infty(f, q) = \mathcal{L}_{\text{align}}(f) + \mathcal{L}_{\text{unif}}(f, q) - 1/t^2
$$

where $\mathcal{L}_{\mathrm{align}}(f) = \mathbb{E}_{x,x^+}\|f(x) - f(x^+)\|^2/2$ and $\mathcal{L}_{\mathrm{unif}}(f, q) = \mathbb{E}_{x \sim p} \log \mathbb{E}_{x^- \sim q} e^{f(x)^\top f(x^-)}$. Here we have used the fact that $f$ lies on the boundary of the hypersphere of radius $1/t$, which gives us the following equivalence between inner products and squared Euclidean norm,

$$2/t^2 - 2f(x)^\top f(x^+) = \|f(x)\|^2 + \|f(x^+)\|^2 - 2f(x)^\top f(x^+) = \|f(x) - f(x^+)\|^2. \quad (9)$$

Taking supremum to obtain $\mathcal{L}_\infty^*(f) = \sup_{q \in \Pi} \mathcal{L}_\infty(f, q)$ we find that the second expression simplifies to,

$$\mathcal{L}_{\mathrm{unif}}^*(f) = \sup_{q \in \Pi} \mathcal{L}_{\mathrm{unif}}(f, q) = \mathbb{E}_{x \sim p} \log \sup_{x^- \nsim x} e^{f(x)^\top f(x^-)} = \mathbb{E}_{x \sim p} \sup_{x^- \nsim x} f(x)^\top f(x^-).$$

Using Eqn. (9), this can be re-expressed as,

$$\mathbb{E}_{x \sim p} \sup_{x^- \nsim x} f(x)^\top f(x^-) = -\mathbb{E}_{x \sim p} \inf_{x^- \nsim x} \|f(x) - f(x^-)\|^2/2 + 1/t^2. \quad (10)$$

The forthcoming theorem exactly characterizes the global optima of $\min_f \mathcal{L}_\infty^*(f)$

**Theorem 7.** *Suppose the downstream task is classification (i.e. $\mathcal{C}$ is finite), and let $\mathcal{L}_\infty^*(f) = \sup_{q \in \Pi} \mathcal{L}_\infty(f, q)$. The infimum $\inf_{f:\ \mathrm{measurable}} \mathcal{L}_\infty^*(f)$ is attained, and any $f^*$ achieving the global minimum is such that $f^*(x) = f^*(x^+)$ almost surely. Furthermore, letting $\mathbf{v}_c = f^*(x)$ for any $x$ such that $h(x) = c$ (so $\mathbf{v}_c$ is well defined up to a set of $x$ of measure zero), $f^*$ is characterized as being any solution to the following ball-packing problem,*

$$\max_{\{\mathbf{v}_c \in \mathbb{S}^{d-1}/t\}_{c \in \mathcal{C}}} \sum_{c \in \mathcal{C}} \rho(c) \cdot \min_{c' \neq c} \|\mathbf{v}_c - \mathbf{v}_{c'}\|^2. \quad (11)$$

*Proof.* Any minimizer of $\mathcal{L}_{\mathrm{align}}(f)$ has the property that $f(x) = f(x^+)$ almost surely. So, in order to prove the first claim, it suffices to show that there exist functions $f \in \arg\inf_f \mathcal{L}_{\mathrm{unif}}^*(f)$ for which $f(x) = f(x^+)$ almost surely. This is because, at that point, we have shown that $\arg\min_f \mathcal{L}_{\mathrm{align}}(f)$ and $\arg\min_f \mathcal{L}_{\mathrm{unif}}^*(f)$ intersect, and therefore any solution of $\mathcal{L}_\infty^*(f) = \mathcal{L}_{\mathrm{align}}(f) + \mathcal{L}_{\mathrm{unif}}^*(f)$ must lie in this intersection.

To this end, suppose that $f \in \arg\min_f \mathcal{L}_{\mathrm{unif}}^*(f)$ but that $f(x) \neq f(x^+)$ with non-zero probability. We shall show that we can construct a new embedding $\hat{f}$ such that $f(x) = f(x^+)$ almost surely, and $\mathcal{L}_{\mathrm{unif}}^*(\hat{f}) \leq \mathcal{L}_{\mathrm{unif}}^*(f)$. Due to Eqn. (10) this last condition is equivalent to showing,

$$\mathbb{E}_{x \sim p} \inf_{x^- \nsim x} \|\hat{f}(x) - \hat{f}(x^-)\|^2 \geq \mathbb{E}_{x \sim p} \inf_{x^- \nsim x} \|f(x) - f(x^-)\|^2. \quad (12)$$

Fix a $c \in \mathcal{C}$, and let $x_c \in \arg\max_{x:h(x)=c} \inf_{x^- \nsim x} \|f(x) - f(x^-)\|^2$. The maximum is guaranteed to be attained, as we explain now. Indeed we know the maximum is attained at some point in the closure $\partial\{x : h(x) = c\} \cup \{x : h(x) = c\}$. Since $\mathcal{X}$ is compact and connected, any point $\bar{x} \in \partial\{x : h(x) = c\} \setminus \{x : h(x) = c\}$ is such that $\inf_{x^- \nsim \bar{x}} \|f(\bar{x}) - f(x^-)\|^2 = 0$ since $\bar{x}$ must belong to $\{x : h(x) = c'\}$ for some other $c'$. Such an $\bar{x}$ cannot be a solution unless all points in $\{x : h(x) = c\}$ also achieve 0, in which case we can simply take $x_c$ to be a point in the interior of $\{x : h(x) = c\}$.

Now, define $\hat{f}(x) = f(x_c)$ for any $x$ such that $h(x) = c$ and $\hat{f}(x) = f(x)$ otherwise. Let us first aim to show that Eqn. (12) holds for this $\hat{f}$. Let us begin to expand the left hand side of Eqn. (12),

$$\mathbb{E}_{x \sim p} \inf_{x^- \nsim x} \|\hat{f}(x) - \hat{f}(x^-)\|^2$$

$$= \mathbb{E}_{\hat{c} \sim \rho} \mathbb{E}_{x \sim p(\cdot|\hat{c})} \inf_{x^- \nsim x} \|\hat{f}(x) - \hat{f}(x^-)\|^2$$

$$= \rho(c) \mathbb{E}_{x \sim p(\cdot|c)} \inf_{x^- \nsim x} \|\hat{f}(x) - \hat{f}(x^-)\|^2$$

$$+ (1 - \rho(c)) \mathbb{E}_{\hat{c} \sim \rho(\cdot|\hat{c} \neq c)} \mathbb{E}_{x \sim p(\cdot|\hat{c})} \inf_{x^- \nsim x} \|\hat{f}(x) - \hat{f}(x^-)\|^2$$

$$= \rho(c) \mathbb{E}_{x \sim p(\cdot|c)} \inf_{x^- \nsim x} \|f(x_c) - f(x^-)\|^2$$

$$+ (1 - \rho(c)) \mathbb{E}_{\hat{c} \sim \rho(\cdot|\hat{c} \neq c)} \mathbb{E}_{x \sim p(\cdot|\hat{c})} \inf_{x^- \nsim x} \|\hat{f}(x) - \hat{f}(x^-)\|^2$$

$$= \rho(c) \inf_{x^- \nsim x_c} \|f(x_c) - f(x^-)\|^2$$

$$+ (1 - \rho(c)) \mathbb{E}_{\hat{c} \sim \rho(\cdot|\hat{c} \neq c)} \mathbb{E}_{x \sim p(\cdot|\hat{c})} \inf_{h(x^-) \neq \hat{c}} \|\hat{f}(x) - \hat{f}(x^-)\|^2 \quad (13)$$

By construction, the first term can be lower bounded by $\inf_{x^- \nsim x_c} \|f(x_c) - f(x^-)\|^2 \geq \mathbb{E}_{x \sim p(\cdot|c)} \inf_{h(x^-) \neq c} \|f(x) - f(x^-)\|^2$ for any $x$ such that $h(x) = c$. To lower bound the second term, consider any fixed $\hat{c} \neq c$ and $x \sim p(\cdot|\hat{c})$ (so $h(x) = \hat{c}$). Define the following two subsets of the input space $\mathcal{X}$

$$\mathcal{A} = \{f(x^-) : f(x^-) \neq \hat{c} \text{ for } x^- \in \mathcal{X}\} \qquad \widehat{\mathcal{A}} = \{f(x^-) \in \mathcal{X} : \hat{f}(x^-) \neq \hat{c} \text{ for } x^- \in \mathcal{X}\}.$$

Since by construction the range of $\hat{f}$ is a subset of the range of $f$, we know that $\widehat{\mathcal{A}} \subseteq \mathcal{A}$. Combining this with the fact that $\hat{f}(x) = f(x)$ whenever $h(x) = \hat{c} \neq c$ we see,

$$\inf_{h(x^-) \neq \hat{c}} \|\hat{f}(x) - \hat{f}(x^-)\|^2 = \inf_{h(x^-) \neq \hat{c}} \|f(x) - \hat{f}(x^-)\|^2$$

$$= \inf_{u \in \widehat{\mathcal{A}}} \|f(x) - u\|^2$$

$$\geq \inf_{u \in \mathcal{A}} \|f(x) - u\|^2$$

$$= \inf_{h(x^-) \neq \hat{c}} \|f(x) - f(x^-)\|^2$$

Using these two lower bounds we may conclude that Eqn. (13) can be lower bounded by,

$$\rho(c) \mathbb{E}_{x \sim p(\cdot|c)} \inf_{h(x^-) \neq c} \|f(x) - f(x^-)\|^2 + (1 - \rho(c)) \mathbb{E}_{\hat{c} \sim \rho(\cdot|\hat{c} \neq c)} \mathbb{E}_{x \sim p(\cdot|\hat{c})} \inf_{h(x^-) \neq \hat{c}} \|f(x) - f(x^-)\|^2$$

which equals $\mathbb{E}_{x \sim p} \inf_{x^- \nsim x} \|f(x) - f(x^-)\|^2$. We have therefore proved Eqn. (12). To summarize the current progress; given an embedding $f$ we have constructed a new embedding $\hat{f}$ that attains lower $\mathcal{L}_{\text{unif}}$ loss and which is constant on $x$ such that $\hat{f}$ is constant on $\{x : h(x) = c\}$. Enumerating $\mathcal{C} = \{c_1, c_2 \ldots, c_{|\mathcal{C}|}\}$, we may repeatedly apply the same argument to construct a sequence of embeddings $f_1, f_2, \ldots, f_{|\mathcal{C}|}$ such that $f_i$ is constant on each of the following sets $\{x : h(x) = c_j\}$ for $j \leq i$. The final embedding in the sequence $f^* = f_{|\mathcal{C}|}$ is such that $\mathcal{L}_{\text{unif}}^*(f^*) \leq \mathcal{L}_{\text{unif}}^*(f)$ and therefore $f^*$ is a minimizer. This embedding is constant on each of $\{x : h(x) = c_j\}$ for $j = 1, 2, \ldots, |\mathcal{C}|$. In other words, $f^*(x) = f^*(x^+)$ almost surely. We have proved the first claim.

Obtaining the second claim is a matter of manipulating $\mathcal{L}_\infty^*(f^*)$. Indeed, we know that $\mathcal{L}_\infty^*(f^*) = \mathcal{L}_{\text{unif}}^*(f^*) - 1/t^2$ and defining $\mathbf{v}_c = f^*(x) = f(x_c)$ for each $c \in \mathcal{C}$, this expression is minimized if and only if $f^*$ attains,

$$\max_f \mathbb{E}_{x \sim p} \inf_{x^- \not\sim x} \|f(x) - f(x^-)\|^2 = \max_f \mathbb{E}_{c \sim \rho} \mathbb{E}_{x \sim p(\cdot|c)} \inf_{h(x^-) \neq c} \|f(x) - f(x^-)\|^2$$

$$= \max_f \sum_{c \in \mathcal{C}} \rho(c) \cdot \inf_{h(x^-) \neq c} \|f(x) - f(x^-)\|^2$$

$$= \max_{\{\mathbf{v}_c \in \mathbb{S}^{d-1}/t\}_{c \in \mathcal{C}}} \sum_{c \in \mathcal{C}} \rho(c) \cdot \min_{c' \neq c} \|\mathbf{v}_c - \mathbf{v}_{c'}\|^2$$

where the final equality inserts $f^*$ as an optimal $f$ and reparameterizes the maximum to be over the set of vectors $\{\mathbf{v}_c \in \mathbb{S}^{d-1}/t\}_{c \in \mathcal{C}}$. $\square$

## A.3 Downstream Generalization

**Theorem 8.** *Suppose $\rho$ is uniform on $\mathcal{C}$ and $f$ is such that $\mathcal{L}_\infty^*(f) - \inf_{\bar{f} \text{ measurable}} \mathcal{L}_\infty^*(\bar{f}) \leq \varepsilon$ with $\varepsilon \leq 1$. Let $\{\mathbf{v}_c^* \in \mathbb{S}^{d-1}/t\}_{c \in \mathcal{C}}$ be a solution to Problem 7, and define $\xi = \min_{c,c^-:c \neq c^-} \|\mathbf{v}_c^* - \mathbf{v}_{c^-}^*\| > 0$. Then there exists a set of vectors $\{\mathbf{v}_c \in \mathbb{S}^{d-1}/t\}_{c \in \mathcal{C}}$ such that the following 1-nearest neighbor classifier,*

$$\hat{h}(x) = \hat{c}, \quad \text{where} \quad \hat{c} = \arg \min_{\bar{c} \in \mathcal{C}} \|f(x) - \mathbf{v}_{\bar{c}}\| \quad \text{(ties broken arbitrarily)}$$

*achieves misclassification risk,*

$$\mathbb{P}(\hat{h}(x) \neq c) \leq \frac{8\varepsilon}{(\xi^2 - 2|\mathcal{C}|\,(1 + 1/t)\varepsilon^{1/2})^2}$$

*Proof.* To begin, using the definition of $\hat{h}$ we know that for any $0 < \delta < \xi$,

$$\mathbb{P}_{x,c}(\hat{h}(x) = c) = \mathbb{P}_{x,c}\left(\|f(x) - \mathbf{v}_c\| \leq \min_{c^-:c^- \neq c} \|f(x) - \mathbf{v}_{c^-}\|\right)$$

$$\geq \mathbb{P}_{x,c}\left(\|f(x) - \mathbf{v}_c\| \leq \delta, \quad \text{and} \quad \delta \leq \min_{c^-:c^- \neq c} \|f(x) - \mathbf{v}_{c^-}\|\right)$$

$$\geq 1 - \mathbb{P}_{x,c}\left(\|f(x) - \mathbf{v}_c\| > \delta\right) - \mathbb{P}_{x,c}\left(\min_{c^-:c^- \neq c} \|f(x) - \mathbf{v}_{c^-}\| < \delta\right)$$

So to prove the result, our goal is now to bound these two probabilities. To do so, we use the bound on the excess risk. Indeed, combining the fact $\mathcal{L}_\infty^*(f) - \inf_{\bar{f} \text{ measurable}} \mathcal{L}_\infty^*(\bar{f}) \leq \varepsilon$ with the notational rearrangements before Theorem 7 we observe that $\mathbb{E}_{x,x^+}\|f(x) - f(x^+)\|^2 \leq 2\varepsilon$.

We have,

$$2\varepsilon \geq \mathbb{E}_{x,x^+}\|f(x) - f(x^+)\|^2 = \mathbb{E}_{c \sim \rho} \mathbb{E}_{x^+ \sim p(\cdot|c)} \mathbb{E}_{x \sim p(\cdot|c)} \|f(x) - f(x^+)\|^2.$$

For fixed $c, x^+$, let $x_c \in \arg \min_{\overline{\{x^+:h(x^+)=c\}}} \mathbb{E}_{x \sim p(\cdot|c)}\|f(x) - f(x^+)\|^2$ where we extend the minimum to be over the closure, a compact set, to guarantee it is attained. Then we have

$$2\varepsilon \geq \mathbb{E}_{c \sim \rho} \mathbb{E}_{x^+ \sim p(\cdot|c)} \mathbb{E}_{x \sim p(\cdot|c)} \|f(x) - f(x^+)\|^2 \geq \mathbb{E}_{c \sim \rho} \mathbb{E}_{x \sim p(\cdot|c)} \|f(x) - \mathbf{v}_c\|^2$$

where we have now defined $\mathbf{v}_c = f(x_c)$ for each $c \in \mathcal{C}$. Note in particular that $\mathbf{v}_c$ lies on the surface of the hypersphere $\mathbb{S}^{d-1}/t$. This enables us to obtain the follow bound using Markov's inequality,

$$\mathbb{P}_{x,c}\left(\|f(x) - \mathbf{v}_c\| > \delta\right) = \mathbb{P}_{x,c}\left(\|f(x) - \mathbf{v}_c\|^2 > \delta^2\right)$$
$$\leq \frac{\mathbb{E}_{x,c}\|f(x) - \mathbf{v}_c\|^2}{\delta^2}$$
$$\leq \frac{2\varepsilon}{\delta^2}.$$

so it remains still to bound $\mathbb{P}_{x,c}\left(\min_{c^-:c^- \neq c}\|f(x) - \mathbf{v}_{c^-}\| < \delta\right)$. Defining $\xi' = \min_{c,c^-:c \neq c^-}\|\mathbf{v}_c - \mathbf{v}_{c^-}\|$, we have the following fact (proven later).

**Fact (see lemma 9):** $\xi' \geq \sqrt{\xi^2 - 2|\mathcal{C}|(1+1/t)\sqrt{\varepsilon}}$.

Using this fact we are able to get control over the tail probability as follows,

$$\mathbb{P}_{x,c}\left(\min_{c^-:c^- \neq c}\|f(x) - \mathbf{v}_{c^-}\| < \delta\right) \leq \mathbb{P}_{x,c}\left(\|f(x) - \mathbf{v}_c\| > \xi' - \delta\right)$$
$$\leq \mathbb{P}_{x,c}\left(\|f(x) - \mathbf{v}_c\| > \xi - \sqrt{\xi^2 - 2|\mathcal{C}|(1+1/t)\varepsilon^{1/2}} - \delta\right)$$
$$= \mathbb{P}_{x,c}\left(\|f(x) - \mathbf{v}_c\|^2 > (\sqrt{\xi^2 - 2|\mathcal{C}|(1+1/t)\varepsilon^{1/2}} - \delta)^2\right)$$
$$\leq \frac{2\varepsilon}{(\sqrt{\xi^2 - 2|\mathcal{C}|(1+1/t)\varepsilon^{1/2}} - \delta)^2}.$$

where this inequality holds for for any $0 \leq \delta \leq \sqrt{\xi^2 - 2|\mathcal{C}|(1+1/t)\varepsilon^{1/2}}$.

Gathering together our tail probability bounds we find that $\mathbb{P}_{x,c}(\hat{h}(x) = c) \geq 1 - \frac{2\varepsilon}{\delta^2} - \frac{2\varepsilon}{(\sqrt{\xi^2 - 2|\mathcal{C}|(1+1/t)\varepsilon^{1/2}} - \delta)^2}$ for any $0 \leq \delta \leq \sqrt{\xi^2 - 2|\mathcal{C}|(1+1/t)\varepsilon^{1/2}}$. That is,

$$\mathbb{P}_{x,c}(\hat{h}(x) \neq c) \leq \frac{2\varepsilon}{\delta^2} + \frac{2\varepsilon}{(\sqrt{\xi^2 - 2|\mathcal{C}|(1+1/t)\varepsilon^{1/2}} - \delta)^2}$$

Since this holds for any $0 \leq \delta \leq \sqrt{\xi^2 - 2|\mathcal{C}|(1+1/t)\varepsilon^{1/2}}$,

$$\mathbb{P}_{x,c}(\hat{h}(x) \neq c) \leq \min_{0 \leq \delta \leq \sqrt{\xi^2 - 2|\mathcal{C}|\varepsilon}}\left\{\frac{2\varepsilon}{\delta^2} + \frac{2\varepsilon}{(\sqrt{\xi^2 - 2|\mathcal{C}|(1+1/t)\varepsilon^{1/2}} - \delta)^2}\right\}.$$

Elementary calculus shows that the minimum is attained at $\delta = \frac{\sqrt{\xi^2 - 2|\mathcal{C}|(1+1/t)\varepsilon^{1/2}}}{2}$. Plugging this in yields the final bound,

$$\mathbb{P}(\hat{h}(x) \neq c) \leq \frac{8\varepsilon}{(\xi^2 - 2|\mathcal{C}|(1+1/t)\varepsilon^{1/2})^2}.$$

$\square$

**Lemma 9.** *Consider the same setting as introduced in Theorem 5. In particular define*

$$\xi' = \min_{c,c^-:c \neq c^-}\|\mathbf{v}_c - \mathbf{v}_{c^-}\|, \qquad \xi = \min_{c,c^-:c \neq c^-}\|\mathbf{v}_c^* - \mathbf{v}_{c^-}^*\|.$$

*where $\{\mathbf{v}_c^* \in \mathbb{S}^{d-1}/t\}_{c \in \mathcal{C}}$ is a solution to Problem 7, and $\{\mathbf{v}_c \in \mathbb{S}^{d-1}/t\}_{c \in \mathcal{C}}$ is defined via $\mathbf{v}_c = f(x_c)$ with $x_c \in \arg\min_{\overline{\{x^+:h(x^+)=c\}}}\mathbb{E}_{x \sim p(\cdot|c)}\|f(x) - f(x^+)\|^2$ for each $c \in \mathcal{C}$. Then we have,*

$$\xi' \geq \sqrt{\xi^2 - 2|\mathcal{C}|(1+1/t)\varepsilon^{1/2}}.$$

*Proof.* Define,

$$X = \min_{c^-:c^- \neq c} \left\| \mathbf{v}_c - \mathbf{v}_{c-} \right\|^2, \qquad X^* = \min_{c^-:c^- \neq c} \left\| \mathbf{v}_c^* - \mathbf{v}_{c-}^* \right\|^2.$$

$X$ and $X^*$ are random due to the randomness of $c \sim \rho$. We can split up the following expectation by conditioning on the event $\{X \leq X^*\}$ and its complement,

$$\mathbb{E}|X - X^*| = \mathbb{P}(X \geq X^*)\mathbb{E}[X - X^*] + \mathbb{P}(X \leq X^*)\mathbb{E}[X^* - X]. \tag{14}$$

Using $\mathcal{L}_\infty^*(f) - \inf_{\bar{f} \text{ measurable}} \mathcal{L}_\infty^*(\bar{f}) \leq \varepsilon$ and the notational re-writing of the objective $\mathcal{L}_\infty^*$ introduced before Theorem 7, we observe the following fact, whose proof we give in a separate lemma after the conclusion of this proof.

**Fact (see lemma 10):** $\mathbb{E}X^* - 2(1 + 1/t)\sqrt{\varepsilon} \leq \mathbb{E}X \leq \mathbb{E}X^*$.

This fact implies in particular $\mathbb{E}[X - X^*] \leq 0$ and $\mathbb{E}[X^* - X] \leq 2(1 + 1/t)\sqrt{\varepsilon}$. Inserting both inequalities into Eqn. 14 we find that $\mathbb{E}|X - X^*| \leq 2(1 + 1/t)\sqrt{\varepsilon}$. In other words, since $\rho$ is uniform,

$$\frac{1}{|\mathcal{C}|} \sum_{c \in \mathcal{C}} \left| \min_{c^-:c^- \neq c} \left\| \mathbf{v}_c - \mathbf{v}_{c-} \right\|^2 - \min_{c^-:c^- \neq c} \left\| \mathbf{v}_c^* - \mathbf{v}_{c-}^* \right\|^2 \right| \leq 2(1 + 1/t)\sqrt{\varepsilon}.$$

From which we can say that for any $c \in \mathcal{C}$,

$$\left| \min_{c^-:c^- \neq c} \left\| \mathbf{v}_c - \mathbf{v}_{c-} \right\|^2 - \min_{c^-:c^- \neq c} \left\| \mathbf{v}_c^* - \mathbf{v}_{c-}^* \right\|^2 \right| \leq 2|\mathcal{C}|(1 + 1/t)\sqrt{\varepsilon}.$$

So $\min_{c^-:c^- \neq c}\|\mathbf{v}_c - \mathbf{v}_{c-}\| \geq \sqrt{\min_{c^-:c^- \neq c}\left\|\mathbf{v}_c^* - \mathbf{v}_{c-}^*\right\|^2 - 2|\mathcal{C}|(1 + 1/t)\varepsilon^{1/2}} \geq \sqrt{\xi^2 - 2|\mathcal{C}|(1 + 1/t)\varepsilon^{1/2}}$. Since this holds for any $c \in \mathcal{C}$, we conclude that $\xi' \geq \sqrt{\xi^2 - 2|\mathcal{C}|(1 + 1/t)\varepsilon^{1/2}}$. $\qquad\square$

**Lemma 10.** *Consider the same setting as introduced in Theorem 5. Define also,*

$$X = \min_{c^-:c^- \neq c} \left\| \mathbf{v}_c - \mathbf{v}_{c-} \right\|^2, \qquad X^* = \min_{c^-:c^- \neq c} \left\| \mathbf{v}_c^* - \mathbf{v}_{c-}^* \right\|^2,$$

*where $\mathbf{v}_c = f(x_c)$ with $x_c \in \arg\min_{\overline{\{x^+:h(x^+)=c\}}} \mathbb{E}_{x \sim p(\cdot|c)}\left\| f(x) - f(x^+)\right\|^2$ for each $c \in \mathcal{C}$. We have,*

$$\mathbb{E}X^* - 2(1 + 1/t)\sqrt{\varepsilon} \leq \mathbb{E}X \leq \mathbb{E}X^*.$$

*Proof.* By Theorem 7 we know there is an $f^*$ attaining the minimum $\inf_{\bar{f} \text{ measurable}} \mathcal{L}_\infty^*(\bar{f})$ and that this $f^*$ attains $\mathcal{L}_{\text{align}}^*(f^*) = 0$, and also minimizes the uniformity term $\mathcal{L}_{\text{unif}}^*(f)$, taking the value $\mathcal{L}_{\text{unif}}^*(f^*) = \mathbb{E}_{c \sim \rho} \max_{c^-:c^- \neq c} \mathbf{v}_{\mathbf{c}}^{*\top} \mathbf{v}_{c-}^*$. Because of this we find,

$$\begin{aligned}
\mathcal{L}_{\text{unif}}^*(f) &\leq \left(\mathcal{L}_\infty^*(f) - \mathcal{L}_\infty^*(f^*)\right) + \left(\mathcal{L}_{\text{align}}^*(f^*) - \mathcal{L}_{\text{align}}^*(f)\right) + \mathcal{L}_{\text{unif}}^*(f^*) \\
&\leq \left(\mathcal{L}_\infty^*(f) - \mathcal{L}_\infty^*(f^*)\right) + \mathcal{L}_{\text{unif}}^*(f^*) \\
&\leq \varepsilon + \mathcal{L}_{\text{unif}}^*(f^*) \\
&= \varepsilon + \mathbb{E}_{c \sim \rho} \max_{c^-:c^- \neq c} \mathbf{v}_{\mathbf{c}}^{*\top} \mathbf{v}_{c-}^*.
\end{aligned}$$

Since we would like to bound $\mathbb{E}_{c\sim\rho}\max_{c^-:c^-\neq c}\mathbf{v_c}^\top\mathbf{v}_{c^-}$ in terms of $\mathbb{E}_{c\sim\rho}\max_{c^-:c^-\neq c}\mathbf{v_c^*}^\top\mathbf{v}_{c^-}^*$, this observation means that is suffices to bound $\mathbb{E}_{c\sim\rho}\max_{c^-:c^-\neq c}\mathbf{v_c}^\top\mathbf{v}_{c^-}$ in terms of $\mathcal{L}_{\text{unif}}^*(f)$. To this end, note that for a fixed $c$, and $x$ such that $h(x) = c$ we have,

$$
\begin{aligned}
\sup_{x^-\nsim x} f(x)^\top f(x^-) &= \sup_{x^-\nsim x}\left\{\mathbf{v}_c^\top f(x^-) + (f(x)-\mathbf{v}_c)^\top f(x^-)\right\}\\
&= \sup_{x^-\nsim x}\mathbf{v}_c^\top f(x^-) - \left\|f(x)-\mathbf{v}_c\right\|/t\\
&\geq \max_{x^-\in\{x_c\}_{c\in\mathcal{C}}}\mathbf{v}_c^\top f(x^-) - \left\|f(x)-\mathbf{v}_c\right\|/t\\
&= \max_{c^-\neq c}\mathbf{v}_c^\top\mathbf{v}_{c^-} - \left\|f(x)-\mathbf{v}_c\right\|/t
\end{aligned}
$$

where the inequality follows since $\{x_c\}_{c\in\mathcal{C}}$ is a subset of the closure of $\{x^- : x^- \nsim x\}$. Taking expectations over $c, x$,

$$
\begin{aligned}
\mathcal{L}_{\text{unif}}^*(f) &= \mathbb{E}_{x,c}\sup_{x^-\nsim x} f(x)^\top f(x^-)\\
&\geq \mathbb{E}_{c\sim\rho}\max_{c^-\neq c}\mathbf{v}_c^\top\mathbf{v}_{c^-} - \mathbb{E}_{x,c}\left\|f(x)-\mathbf{v}_c\right\|/t\\
&\geq \mathbb{E}_{c\sim\rho}\max_{c^-\neq c}\mathbf{v}_c^\top\mathbf{v}_{c^-} - \sqrt{\mathbb{E}_{x,c}\left\|f(x)-\mathbf{v}_c\right\|^2}/t\\
&\geq \mathbb{E}_{c\sim\rho}\max_{c^-\neq c}\mathbf{v}_c^\top\mathbf{v}_{c^-} - \sqrt{\varepsilon}/t.
\end{aligned}
$$

So since $\varepsilon \leq \sqrt{\varepsilon}$, we have found that

$$
\mathbb{E}_{c\sim\rho}\max_{c^-\neq c}\mathbf{v}_c^\top\mathbf{v}_{c^-} \leq \sqrt{\varepsilon}/t + \varepsilon + \mathbb{E}_{c\sim\rho}\max_{c^-:c^-\neq c}\mathbf{v_c^*}^\top\mathbf{v}_{c^-}^* \leq (1+1/t)\sqrt{\varepsilon} + \mathbb{E}_{c\sim\rho}\max_{c^-:c^-\neq c}\mathbf{v_c^*}^\top\mathbf{v}_{c^-}^*.
$$

Of course we also have,

$$
\mathbb{E}_{c\sim\rho}\max_{c^-:c^-\neq c}\mathbf{v_c^*}^\top\mathbf{v}_{c^-}^* = \mathcal{L}_{\text{unif}}^*(f^*) \leq \mathbb{E}_{c\sim\rho}\max_{c^-:c^-\neq c}\mathbf{v}_c^\top\mathbf{v}_{c^-}
$$

since the embedding $f(x) = \mathbf{v}_c$ whenever $h(x) = c$ is also a feasible solution. Combining these two inequalities with the simple identity $\mathbf{x}^\top\mathbf{y} = 1/t^2 - \left\|\mathbf{x}-\mathbf{y}\right\|^2/2$ for all length $1/t$ vectors $\mathbf{x}, \mathbf{y}$, we find,

$$
\begin{aligned}
1/t^2 - \mathbb{E}_{c\sim\rho}\max_{c^-:c^-\neq c}\left\|\mathbf{v_c^*}-\mathbf{v}_{c^-}^*\right\|^2/2 &\leq 1/t^2 - \mathbb{E}_{c\sim\rho}\max_{c^-:c^-\neq c}\left\|\mathbf{v_c}-\mathbf{v}_{c^-}\right\|^2/2\\
&\leq 1/t^2 - \mathbb{E}_{c\sim\rho}\max_{c^-:c^-\neq c}\left\|\mathbf{v_c^*}-\mathbf{v}_{c^-}^*\right\|^2/2 + (1+1/t)\sqrt{\varepsilon}.
\end{aligned}
$$

Subtracting $1/t^2$ and multiplying by $-2$ yields the result.

$\square$

## B   GRAPH REPRESENTATION LEARNING

We describe in detail the hard sampling method for graphs whose results are reported in Section 5.2. Before getting that point, in the interests of completeness we cover some required background details on the InfoGraph method of Sun et al. (2020). For further information see the original paper (Sun et al., 2020).

### B.1 BACKGROUND ON GRAPH REPRESENTATIONS

We observe a set of graphs $\mathbf{G} = \{G_j \in \mathbb{G}\}_{j=1}^{n}$ sampled according to a distribution $p$ over an ambient graph space $\mathbb{G}$. Each node $u$ in a graph $G$ is assumed to have features $h_u^{(0)}$ living in some Euclidean space. We consider a $K$-layer graph neural network, whose $k$-th layer iteratively computes updated embeddings for each node $v \in G$ in the following way,

$$h_v^{(k)} = \text{COMBINE}^{(k)} \left( h_v^{(k-1)}, \text{AGGREGATE}^{(k)} \left( \left\{ \left( h_v^{(k-1)}, h_u^{(k-1)}, e_{uv} \right) : u \in \mathcal{N}(v) \right\} \right) \right)$$

where $\text{COMBINE}^{(k)}$ and $\text{AGGREGATE}^{(k)}$ are parameterized learnable functions and $\mathcal{N}(v)$ denotes the set of neighboring nodes of $v$. The $K$ embeddings for a node $u$ are collected together to obtain a single final summary embedding for $u$. As recommended by Xu et al. (2019) we use concatenation, $h^u = h^u(G) = \text{CONCAT} \left( \{h_u^{(k)}\}_{k=1}^{K} \right)$ to obtain an embedding in $\mathbb{R}^d$. Finally, the node representations are combined together into a length $d$ graph level embedding using a readout function,

$$H(G) = \text{READOUT} \left( \{h^u\}_{u \in G} \right)$$

which is typically taken to be a simple permutation invariant function such as the sum or mean. The InfoGraph method aims to maximize the mutual information between the graph level embedding $H(G)$ and patch-level embeddings $h^u(G)$ using the following objective,

$$\max_h \mathbb{E}_{G \sim p} \frac{1}{|G|} \sum_{u \in G} I \left( h^u(G); H(G) \right)$$

In practice the population distribution $p$ is replaced by its empirical counterpart, and the mutual information $I$ is replaced by a variational approximation $I_T$. In line with Sun et al. (2020) we use the Jensen-Shannon mutual information estimator as formulated by Nowozin et al. (2016). It is defined using a neural network discriminator $T : \mathbb{R}^{2d} \to \mathbb{R}$ as,

$$I_T \left( h^u(G); H(G) \right) = \mathbb{E}_{G \sim p} \left[ -\text{sp}(-T \left( h^u(G), H(G) \right)) \right] - \mathbb{E}_{(G,G') \sim p \times p} \left[ \text{sp}(T \left( h^u(G), H(G') \right)) \right]$$

where $\text{sp}(z) = \log(1 + e^z)$ denotes the softplus function. The finial objective is the joint maximization over $h$ and $T$,

$$\max_{\theta, \psi} \mathbb{E}_{G \sim p} \frac{1}{|G|} \sum_{u \in G} I_T \left( h^u(G); H(G) \right)$$

### B.2 HARD NEGATIVE SAMPLING FOR LEARNING GRAPH REPRESENTATIONS

In order to derive a simple modification of the NCE hard sampling technique that is appropriate for use with InfoGraph, we first provide a mildly generalized view of hard sampling. Recall that the NCE contrastive objective can be decomposed into two constituent pieces,

$$\mathcal{L}(f, q) = \mathcal{L}_{\text{align}}(f) + \mathcal{L}_{\text{unif}}(f, q)$$

where $q$ is in fact a family of distributions $q(x^-; x)$ over $x^-$ that is indexed by the possible values of the anchor $x$. $\mathcal{L}_{\text{align}}$ performs the role of "aligning" positive pairs (embedding near to one-another), while $\mathcal{L}_{\text{unif}}$ repels negative pairs. The hard sampling framework aims to solve,

$$\inf_f \sup_q \mathcal{L}(f, q).$$

In the case of NCE loss we take,

$$\mathcal{L}_{\text{align}}(f) = -\mathbb{E}_{\substack{x \sim p \\ x^+ \sim p_x^+}} f(x)^T f(x^+),$$

$$\mathcal{L}_{\text{unif}}(f, q) = \mathbb{E}_{\substack{x \sim p \\ x^+ \sim p_x^+}} \log \left\{ e^{f(x)^T f(x^+)} + Q \mathbb{E}_{x^- \sim q}[e^{f(x)^T f(x^-)}] \right\}.$$

View this view, we can easily adapt to the InfoGraph framework, taking

$$\mathcal{L}_{\text{align}}(h, T) = -\mathbb{E}_{G \sim p} \frac{1}{|G|} \sum_{u \in G} \text{sp}(-T(h^u(G), H(G))),$$

$$\mathcal{L}_{\text{unif}}(h, T, q) = -\mathbb{E}_{G \sim p} \frac{1}{|G|} \sum_{u \in G} \mathbb{E}_{G' \sim q} \text{sp}(T(h^u(G), H(G')))$$

Denote by $\hat{p}$ the distribution over nodes $u \in \mathbb{R}^s$ defined by first sampling $G \sim p$, then sampling $u \in G$ uniformly over all nodes of $G$. Then these two terms can be simplified to

$$\mathcal{L}_{\text{align}}(h, T) = -\mathbb{E}_{u \sim \hat{p}} \text{sp}(-T(h^u(G), H(G))),$$

$$\mathcal{L}_{\text{unif}}(h, T, q) = -\mathbb{E}_{(u, G') \sim \hat{p} \times q} \text{sp}(T(h^u(G), H(G')))$$

At this point it becomes clear that, just as with NCE, a distribution $q^* \in \arg\max_q \mathcal{L}(f, q)$ in the InfoGraph framework if it is supported on $\arg\max_{G' \in \mathbb{G}} \text{sp}(T(h^u(G), H(G')))$. Although this is still hard to compute exactly, it can be approximated by,

$$q_u^\beta(G') \propto \exp\left(\beta T(h^u(G), H(G))\right) \cdot p(G').$$

## C ADDITIONAL EXPERIMENTS

### C.1 HARD NEGATIVES WITH LARGE BATCH SIZES

The vision experiments in the main body of the paper are all based off the SimCLR framework (Chen et al., 2020a). They use a relatively small batch size (up to 512). In order to test whether our hard negatives sampling method can help when the negative batch size is very large, we also run experiments using MoCo-v2 with standard negative memory bank size $N = 65536$ (He et al., 2020; Chen et al., 2020c). We adopt the official MoCo-v2 code[2]. Embeddings are trained for 200 epochs, with batch size 128. Figure 6 summarizes the results. We find that hard negative sampling can still improve the generalization of embeddings trained on CIFAR10: MoCo-v2 attains linear readout accuracy of 88.08%, and MoCo-v2 with hard negatives ($\beta = 0.2$, $\tau^+ = 0$) attains 88.47%.

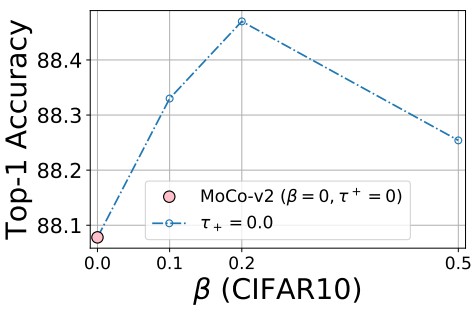

Figure 6: Hard negative sampling using MoCo-v2 framework. Results show that hard negative samples can still be useful when the negative memory bank is very large (in this case $N = 65536$).

### C.2 ABLATIONS

[2]https://github.com/facebookresearch/moco

To study the affect of varying the concentration parameter $\beta$ on the learned embeddings Figure 9 plots cosine similarity histograms of pairs of similar and dissimilar points. The results show that for $\beta$ moving from 0 through 0.5 to 2 causes both the positive and negative similarities to gradually skew left. In terms of downstream classification, an important property is the *relative* difference in similarity between positive and negative pairs. In this case $\beta = 0.5$ find the best balance (since it achieves the highest downstream accuracy). When $\beta$ is taken very large ($\beta = 6$), we see a change in conditions. Both positive and negative pairs are assigned higher similarities in general. Visually it seems that the positive and negative histograms for $\beta = 6$ overlap a lot more than for smaller values, which helps explain why the linear readout accuracy is lower for $\beta = 6$.

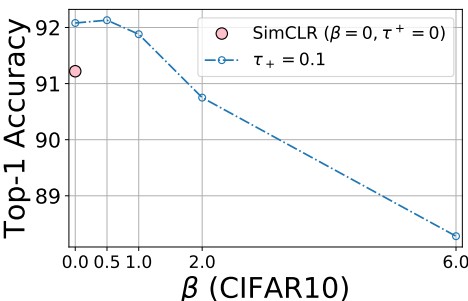

Figure 7: The effect of varying concentration parameter $\beta$ on linear readout accuracy for CIFAR10. (Complements the left and middle plot from Figure 4.)

Figure 12 gives real examples of hard vs. uniformly sampled negatives. Given an anchor $x$ (a monkey) and trained embedding $f$ (trained on STL10 using standard SimCLR for 400 epochs), we sample a batch of 128 images. The top row shows the ten negatives $x^-$ that have the largest inner product $f(x)^\top f(x^-)$, while the bottom row is a random sample from from the same batch. Negatives with the largest inner product with the anchor correspond to the items in the batch are the most important terms in the objective since they are given the highest weighting by $q_\beta^-$. Figure 12 shows that "real" hard negatives are conceptually similar to the idea as proposed in Figure 1: hard negatives are semantically similar to the anchor, possessing various similarities, including color (browns and greens), texture (fur), and objects (animals vs machinery).

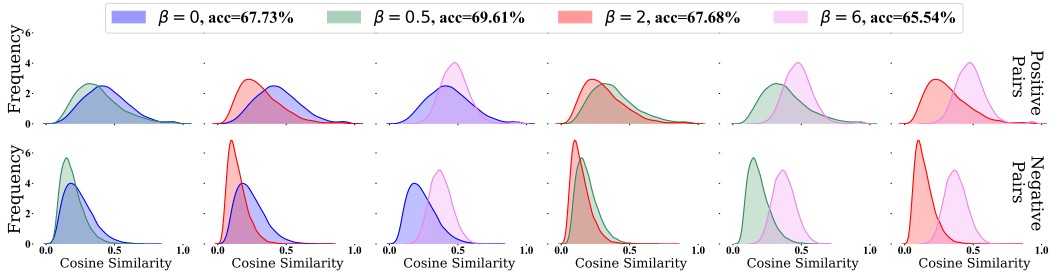

Figure 8: Histograms of cosine similarity of pairs of points with different label (bottom) and same label (top) for embeddings trained on CIFAR100 with different values of $\beta$. Histograms overlaid pairwise to allow for easy comparison.

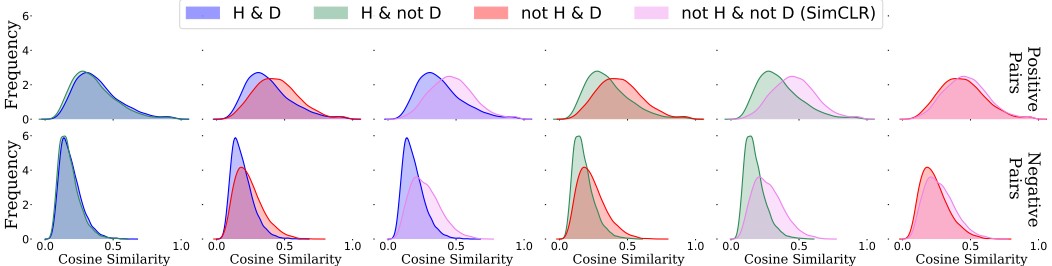

Figure 9: Histograms of cosine similarity of pairs of points with the same label (top) and different labels (bottom) for embeddings trained on CIFAR100 with four different objectives. H=Hard Sampling, D=Debiasing. Histograms overlaid pairwise to allow for convenient comparison.

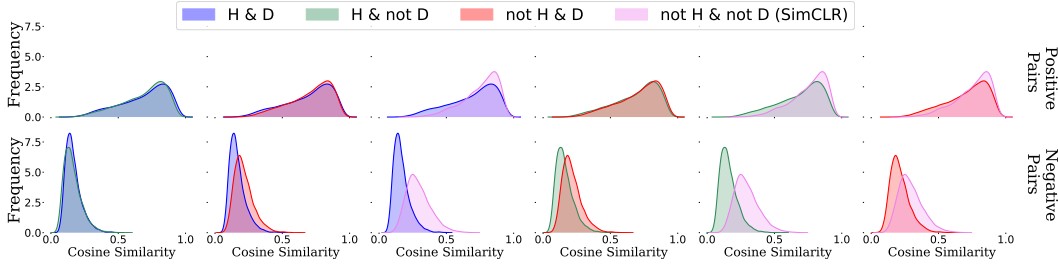

Figure 10: Histograms of cosine similarity of pairs of points with the same label (top) and different labels (bottom) for embeddings trained on CIFAR10 with four different objectives. H=Hard Sampling, D=Debiasing. Histograms overlaid pairwise to allow for convenient comparison.

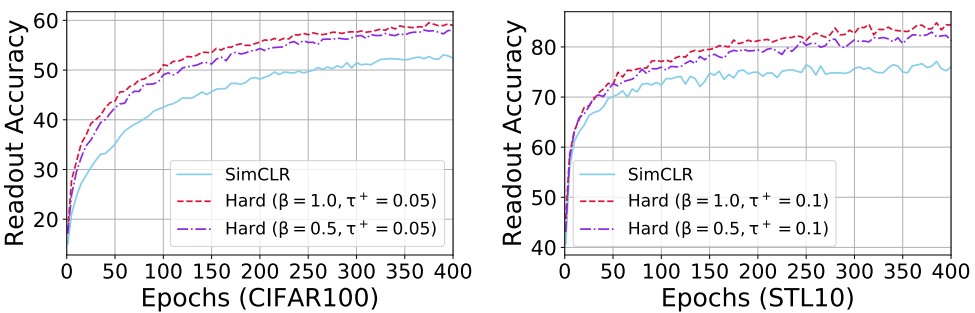

Figure 11: Hard sampling takes much fewer epochs to reach the same accuracy as SimCLR does in 400 epochs; for STL10 with $\beta = 1$ it takes only 60 epochs, and on CIFAR100 it takes 125 epochs (also with $\beta = 1$).

## D EXPERIMENTAL DETAILS

Figure 13 shows PyTorch-style pseudocode for the standard objective, the debiased objective, and the hard sampling objective. The proposed hard-sample loss is very simple to implement, requiring only two extra lines of code compared to the standard objective.

### D.1 VISUAL REPRESENTATIONS

We implement SimCLR in PyTorch. We use a ResNet-50 (He et al., 2016) as the backbone with embedding dimension 2048 (the representation used for linear readout), and projection head into the lower 128-dimensional space (the embedding used in the contrastive objective). We use the Adam optimizer (Kingma & Ba, 2015) with learning rate 0.001 and weight decay $10^{-6}$. Code available at https://github.com/joshr17/HCL. Since we adopt the SimCLR framework, the number of negative samples $N = 2(\text{batch size} - 1)$. Since we always take the batch size to be a power of 2 $(16, 32, 64, 128, 256)$ the negative batch sizes are $30, 62, 126, 254, 510$ respectively. Unless otherwise stated, all models are trained for 400 epochs.

**Annealing $\beta$ Method:** We detail the annealing method whose results are given in Figure 4. The idea is to reduce the concentration parameter down to zero as training progresses. Specifically, suppose we have $e$ number of total training epochs. We also specify a number $\ell$ of "changes" to the concentration parameter we shall make. We initialize the concentration parameter $\beta_1 = \beta$ (where this $\beta$ is the number reported in Figure 4), then once every $e/\ell$ epochs we reduce $\beta_i$ by $\beta/\ell$. In other words, if we are currently on $\beta_i$, then $\beta_{i+1} = \beta_i - \beta/\ell$, and we switch from $\beta_i$ to $\beta_{i+1}$ in epoch number $i \cdot e/\ell$. The idea of this method is to select particularly difficult negative samples early on order to obtain useful gradient information early on, but later (once the embedding is already quite good) we reduce the "hardness" level so as to reduce the harmful effect of only approximately correcting for false negatives (negatives with the same labels as the anchor).

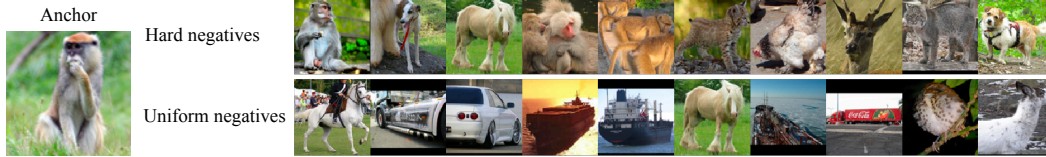

Figure 12: Qualitative comparison of hard negatives and uniformly sampled negatives for embedding trained on STL10 for 400 epochs using SimCLR. Top row: selecting the 10 images with highest inner product with anchor in latent space from a batch of 128 inputs. Bottom row: a set of random samples from the same batch. Hard negatives are semantically much more similar to the anchor than uniformly sampled negatives - hard negatives possess many similar characteristics to the anchor, including texture, colors, animals vs machinery.

```python
# pos     : exp of inner products for positive examples
# neg     : exp of inner products for negative examples
# N       : number of negative examples
# t       : temperature scaling
# tau_plus: class probability
# beta    : concentration parameter

#Original objective
standard_loss = -log(pos.sum() / (pos.sum() + neg.sum()))

#Debiased objective
Neg = max((-N*tau_plus*pos + neg).sum() / (1-tau_plus), e**(-1/t))
debiased_loss = -log(pos.sum() / (pos.sum() + Neg))

#Hard sampling objective (Ours)
reweight = (beta*neg) / neg.mean()
Neg = max((-N*tau_plus*pos + reweight*neg).sum() / (1-tau_plus), e**(-1/t))
hard_loss = -log( pos.sum() / (pos.sum() + Neg))
```

Figure 13: Pseudocode for our proposed new hard sample objective, as well as the original NCE contrastive objective, and debiased contrastive objective. In each case we take the number of positive samples to be $M = 1$. The implementation of our hard sampling method only requires two additional lines of code compared to the standard objective.

We also found the annealing in the opposite direction ("down") achieved similar performance.

**Bias-variance of empirical estimates in hard-negative objective:**    Recall the final hard negative samples objective we derive is,

$$\mathop{\mathbb{E}}_{\substack{x \sim p \\ x^+ \sim p_x^+}} \left[ -\log \frac{e^{f(x)^T f(x^+)}}{e^{f(x)^T f(x^+)} + \frac{Q}{\tau^-}\left(\mathbb{E}_{x^- \sim q_\beta}[e^{f(x)^T f(x^-)}] - \tau^+ \mathbb{E}_{v \sim q_\beta^+}[e^{f(x)^T f(v)}]\right)} \right]. \tag{15}$$

This objective admits a practical counterpart by using empirical approximations to $\mathbb{E}_{x^- \sim q_\beta}[e^{f(x)^T f(x^-)}]$ and $\mathbb{E}_{v \sim q_\beta^+}[e^{f(x)^T f(v)}]$. In practice we use a fairly large number of samples (e.g. $N = 510$) to approximate the first expectation, and only $M = 1$ samples to approximate the second. Clearly in both cases the resulting estimator is unbiased. Further, since the first expectation is approximated using many samples, and the integrand is bounded, the resulting estimator is well concentrated (e.g. apply Hoeffding's inequality out-of-the-box). But what about the second expectation? This might seem uncontrolled since we use only one sample, however it turns out that the random variable $X = e^{f(x)^T f(v)}$ where $x \sim p$ and $v \sim q_\beta^+$ has variance that is bounded by $\mathcal{L}_{\text{align}}(f)$.

**Lemma 11.** *Consider the random variable $X = e^{f(x)^T f(v)}$ where $x \sim p$ and $v \sim q_\beta^+$. Then* $Var(X) \leq \mathcal{O}(\mathcal{L}_{align}(f))$.

Recall that $\mathcal{L}_{\text{align}}(f) = \mathbb{E}_{x,x^+}\|f(x) - f(x^+)\|^2/2$ is termed *alignment*, and Wang & Isola (2020) show that the contrastive objective jointly optimize *alignment* and *uniformity*. Lemma 11 therefore shows that as training evolves, the variance of the $X = e^{f(x)^T f(v)}$ where $x \sim p$ and $v \sim q_\beta^+$ is bounded by a term that we expect to see becoming small, suggesting that using a single sample ($M = 1$) to approximate this expectation is not unreasonable. We cannot, however, say more than this since we have no guarantee that $\mathcal{L}_{\text{align}}(f)$ goes to zero.

*Proof.* Fix an $x$ and recall that we are considering $q_\beta^+(\cdot) = q_\beta^+(\cdot; x)$. First let $X'$ be an i.i.d. copy of $X$, and note that, conditioning on $x$, we have $2\text{Var}(X|x) = \text{Var}(X|x) + \text{Var}(X'|x) = \text{Var}(X - X'|x) \leq \mathbb{E}\big[(X - X')^2|x\big]$. Bounding this difference,

$$
\begin{aligned}
\mathbb{E}\big[(X - X')^2|x\big] &= \mathbb{E}_{v,v'\sim q_\beta^+}\left(e^{f(x)^\top f(v)} - e^{f(x)^\top f(v')}\right)^2 \\
&\leq \mathbb{E}_{v,v'\sim q_\beta^+}\left(e^{1/t^2}\big[f(x)^\top f(v) - f(x)^\top f(v')\big]\right)^2 \\
&\leq e^{1/t^4}\mathbb{E}_{v,v'\sim q_\beta^+}\left(\big[\|f(x)\|\|f(v) - f(v')\|\big]\right)^2 \\
&= \frac{e^{1/t^4}}{t^2}\mathbb{E}_{v,v'\sim q_\beta^+}\|f(v) - f(v')\|^2 \\
&\leq \mathcal{O}\left(\mathbb{E}_{v,v'\sim p^+}\|f(v) - f(v')\|^2\right)
\end{aligned}
$$

where the first inequality follows since $f$ lies on the sphere of radius $1/t$, the second inequality by Cauchy–Schwarz, the third again since $f$ lies on the sphere of radius $1/t$, and the fourth since $q_\beta^+$ is absolutely continuous with respect to $p^+$ with bounded ratio.

Since $p^+(x^+) = p(x^+|h(x))$ only depends on $c = h(x)$, rather than $x$ itself, taking expectations over $x \sim p$ is equivalent to taking expectations over $c \sim \rho$. Further, $\rho(c)p(v|c)p(v'|c) = p(v)p(v'|c) = p(v)p_v^+(v')$. So $\mathbb{E}_{c\sim\rho}\mathbb{E}_{v,v'\sim p^+}\|f(v) - f(v')\|^2 = \mathbb{E}_{x,x^+}\|f(x) - f(x^+)\|^2 = 2\mathcal{L}_{\text{align}}(f)$, where $x \sim p$ and $x^+ \sim p_x^+$. Thus we obtain the lemma. $\square$

```
1  # pos     : exp of inner products for positive examples
2  # neg     : exp of inner products for negative examples
3  # N       : number of negative examples
4  # t       : temperature scaling
5  # tau_plus: class probability
6  # beta    : concentration parameter
7
8  #Clipping negatives trick before computing reweighting
9  reweight = 2*neg / max( neg.max().abs(), neg.min().abs() )
10 reweight = (beta*reweight) / reweight.mean()
11 Neg = max((-N*tau_plus*pos + reweight*neg).sum() / (1-tau_plus), e**(-1/t))
12 hard_loss = -log( pos.sum() / (pos.sum() + Neg))
```

Figure 14: In cases where the learned embedding is not normalized to lie on a hypersphere we found that clipping the negatives to live in a fixed range (in this case $[-2, 2]$) stabilizes optimization.

## D.2    GRAPH REPRESENTATIONS

All datasets we benchmark on can be downloaded at www.graphlearning.io from the TU-Dataset repository of graph classification problems (Morris et al., 2020). Information on basic statistics of the datasets is included in Tables 3 and 4. For fair comparison to the original InfoGraph method, we adopt the official code, which can be found at https://github.com/

`fanyun-sun/InfoGraph`. We modify only the `gan_losses.py` script, adding in our proposed hard sampling via reweighting. For simplicity we trained all models using the same set of hyperparameters: we used the GIN architecture (Xu et al., 2019) with $K = 3$ layers and embedding dimension $d = 32$. Each model is trained for 200 epochs with batch size 128 using the Adam optimizer (Kingma & Ba, 2015). with learning rate 0.001, and weight decay of $10^{-6}$. Each embedding is evaluated using the average accuracy 10-fold cross-validation using an SVM as the classifier (in line with the approach taken by Morris et al. (2020)). Each experiment is repeated from scratch 10 times, and the distribution of results from these 10 runs is plotted in Figure 3.

Since the graph embeddings are not constrained to lie on a hypersphere, for a batch we clip all the inner products to live in the interval $[-2, 2]$ while computing the reweighting, as illustrated in Figure 14. We found this to be important for stabilizing optimization.

| Dataset | DD | PTC | REDDIT-B | PROTEINS |
|---|---|---|---|---|
| No. graphs | 1178 | 344 | 2000 | 1113 |
| No. classes | 2 | 2 | 2 | 2 |
| Avg. nodes | 284.32 | 14.29 | 429.63 | 39.06 |
| Avg. Edges | 715.66 | 14.69 | 497.75 | 72.82 |

Table 3: Basic statistics for graph datasets.

| Dataset | ENZYMES | MUTAG | IMDB-B | IMDB-M |
|---|---|---|---|---|
| No. graphs | 600 | 188 | 1000 | 1500 |
| No. classes | 6 | 2 | 2 | 3 |
| Avg. nodes | 32.63 | 17.93 | 19.77 | 13.00 |
| Avg. Edges | 62.14 | 19.79 | 96.53 | 65.94 |

Table 4: Basic statistics for graph datasets.

### D.3 SENTENCE REPRESENTATIONS

We adopt the official quick-thoughts vectors experimental settings, which can be found at `https://github.com/lajanugen/S2V`. We keep all hyperparameters at the default values and change only the `s2v-model.py` script. Since the official BookCorpus dataset Kiros et al. (2015) is not available, we use an unofficial version obtained using the following repository: `https://github.com/soskek/bookcorpus`. Since the sentence embeddings are also not constrained to lie on a hypersphere, we use the same clipping trick as for the graph embeddings, illustrated in Figure 14.

After training on the BookCorpus dataset, we evaluate the embeddings on six different classification tasks: paraphrase identification (MSRP) (Dolan et al., 2004), question type classification (TREC) (Voorhees & Harman, 2002), opinion polarity (MPQA) (Wiebe et al., 2005), subjectivity classification (SUBJ) (Pang & Lee, 2004), product reviews (CR) (Hu & Liu, 2004), and sentiment of movie reviews (MR) (Pang & Lee, 2005).

## E FURTHER DISCUSSION

**Comparison with Kalantidis et al. (2020):** Kalantidis et al. (2020) also consider ways to sample negatives, and propose a mixing strategy for hard negatives, called MoCHi. The main points of difference are: 1) MoCHi considers the benefit of hard negatives, but does not consider the possibility of false negatives (Principle 1), which we found to be valuable. 2) MoCHi introduces three extra hyperparameters, while our method introduces only two $(\beta, \tau^+)$. If we discard Principle 1 (i.e. $\tau^+$) then only $\beta$ requires tuning. 3) our method introduces zero computational overhead by utilizing within-batch reweighting, whereas MoCHi involves a small amount of extra computation.

**Limitations of proposed method.** There are two main sources of approximation in the hard negative sampling algorithm we propose to use in practice. First, our hard negatives objective requires computing an expectation over $p_x^+(x^+)$, the distributions on points with the same $x^+$ class label as $x$.

Due to lack of supervision, we are not able to sample exactly from $p_x^+$. To circumnavigate this, we propose using samples generated using data augmentation, which we find works well in practice, but deviates from the original formulation and framework. We find this approximation has important implications for practitioners, since we observe that when using oracle access to $p_x^+$ the downstream performance improves monotonically as the concentration parameter $\beta$ increases (see Fig. 4, middle), while when using the practical approximation, large values of $\beta$ start to hurt performance, thereby requiring that $\beta$ be tuned (values in the range $(0.5, 2)$ tend to work well in general). Second, we estimate the expectation $\mathbb{E}_{v \sim q_\beta^+}[e^{f(x)^\top f(v)}]$ using just a single (importance weighted) sample. This is done in the interests of efficiency – since using $M > 1$ samples significantly increases the effective batch size (this is the more costly multi-view setting), increasing memory and runtime costs. We anticipate that using $M > 1$ would improve performance further, in exchange for this extra price, but leave experimentation to future work. That said, our analysis finds that the variance of the variable $e^{f(x)^\top f(v)}$ with $v \sim q_\beta^+$ is controlled by the alignment objective, as shown in Lemma 11, which suggests that in later stage training the variance of the single-sample estimator may be reasonably small.

