# OpenReview forum: "Contrastive Learning with Hard Negative Samples"
_ICLR.cc/2021/Conference — ICLR 2021 Poster_

### Official Review · AnonReviewer4 · 2020-10-24
**Good Submission**

**Rating:** 7
**Confidence:** 4

**Review:**

Summary:

This paper investigated how to sample informative/hard negative examples for self-supervised contrastive learning without label information. To tackle this challenge, this paper proposed an efficient tunable sampling distribution to select negative samples that are similar to the query when the true label or similarity information is not accessible. Positive-unlabeled learning is used to address the challenge of no label, while the importance sampling technique is used for efficient sampling.

My main concerns are (1) how to distinguish hard negative examples and same-class samples as Fig.1 depicts, (2) how to sample $v$ in Eq.(4) to estimate the expectation.

Pros:

\+ This paper is the first to propose a hard negative sampling for unsupervised contrastive representation learning.

\+ This paper provides a theoretical analysis of the proposed method that can tightly cluster similar inputs while departing the clusters from each other.

\+ The proposed method can be easily implemented with few additional lines of code.

\+ Experiments are conducted on several datasets (STL10, CIFAR100, CIFAR10). The proposed method works well even with a small number of negative samples.

\+ This paper has a high writing quality. The paper is clearly written and well organized. The motivation, challenges, and contributions are clearly stated.

\+ This paper shows plenty of visualized results to intuitively and detailedly show how the proposed method works.

Cons:

\- The experiments are conducted only on small datasets. Experiments on larger datasets, including Imagenet-1k and Imagenet-100 are not provided, especially the latter. The most related work, debiased [1], has conducted experiments on Imagenet-100.


Questions:

It is not clear to me how to distinguish hard negative samples and same-class samples. Taking Figure 1 for example, how to distinguish 'oak' from other types of trees without labels?

In Sec. 3.1, $\beta$ is used to meet Principle 2. Since $\beta $ is like the temperature in softmax for scaling and does not change the order/rank of $f(x)^Tf(x^-)$, it is not clear how using $\beta$ satisfies Principle 2.

As shown in Eq.(4), estimating $E_{v\sim q_{\beta}^+}$ requires sampling $v$ from $q_{\beta}^+$. Are there any comments or ablations on the number of samples like Fig. 4(c) in [1]?

The paper focuses on hard negative samples mining with a relatively small number of negative samples (i.e., less than 512). Since memory-bank/queue-based methods like MoCo [2] has a relatively large number of negative samples (i.e., 65536), is it possible to improve the performance of MoCo, or reduce the number of negative samples?

References:

[1] Ching-Yao Chuang, Joshua Robinson, Lin Yen-Chen, Antonio Torralba, and Stefanie Jegelka. De- biased Contrastive Learning. In Advances in Neural Information Processing Systems (NeurIPS), 2020.
[2] Kaiming He, Haoqi Fan, Yuxin Wu, Saining Xie, and Ross Girshick. Momentum contrast for unsupervised visual representation learning. In IEEE Conference on Computer Vision and Pattern Recognition (CVPR), pp. 9729–9738, 2020.










*******************************************

Final decision:

I would keep my score unchanged.

As for Principle 1, the authors said that upholding Principle 1 is impossible with no supervision, and they proposed to uphold Principle 1 approximately. This is acceptable to me as they build on ideas from positive-unlabeled learning.

This paper is clearly written and well organized. Both empirical and theoretical analysis is provided. The feedback addressed my concerns well.

---

> ### Author Response · Authors · 2020-11-19
> **Thank you for your thoughtful review and encouraging comments**
>
> We would like to address your main questions:
>
> > It is not clear to me how to distinguish hard negative samples and same-class samples
>
> The formalism underlying our method requires that a set of classes is specified. In this way we specify the difference between same-class and hard-negative (of different class, but nearby in latent space). This does not affect the practical method except for informing the choice of $\tau^+$.
>
> > $\beta$ ... does not change the order $f(x)^\top f(x^-)$
>
> Indeed, this is correct: the ordering of “hardest” to least hard negatives is fixed. However, varying beta controls how much weight is placed on the “harder” examples. Taking large $\beta$ will weigh the largest inner product the highest, and give almost 0 weighting to all others. In this way taking beta large focuses on only the hardest samples.
>
>
> > Are there any comments or ablations on the number of [positive] samples?
>
> We think it is a good idea to study the variance of these empirical estimates. It is not immediately clear if the empirical estimate using only $M=1$ positive sample is well concentrated. However, we provide a new lemma (Lemma 11, appendix) which shows that the variance of the random variable $X=e^{f(x)^\top f(v)}$ where $x \sim p$ and $v \sim q_\beta^+$ is bounded by $\mathcal O( \mathcal L_\text{align}(f))$ (where $\mathcal L_\text{align}$ is the alignment loss from Wang & Isola ICML 2020). Since Wang & Isola show that the contrastive objective optimizes $\mathcal  L_\text{align}(f) + \mathcal  L_\text{unif}(f)$, we expect $\mathcal  L_\text{align}(f)$ to be small, meaning the random variable $X$ has low variance.
>
> > Is it possible to improve the performance of MoCo?
>
> Yes. See Figure 6, appendix (updated version of paper). Based on your suggestion we have trained embeddings using MoCo-v2 with hard sampling on CIFAR10 for 200 epochs with $\tau^+=0$ and $\beta \in [0.1,0.2,0.5]$  and the standard negative memory bank size $N=65536$. We find that  linear readout accuracy improves in all cases. Standard MoCo-v2 achieves 88.08%, while MoCo-v2 + hard negatives  (with $\beta=0.2$) improves accuracy to 88.47%. Scores are relatively low since we only had time to train for 200 epochs so far. We are now training for more epochs, and tuning $\beta$ and $\tau^+$ and will include full results in an updated version of the paper once complete.

---

### Official Review · AnonReviewer1 · 2020-10-28
**Official Blind Review #1**

**Rating:** 7
**Confidence:** 3

**Review:**

The paper proposes a novel noise contrastive estimation (NCE) objective that incorporates hard-negative samples without similarity supervision, e.g., assuming unsupervised learning. To this end, it modifies the denominator of the original NCE by (a) re-weighting the negative samples based on the euclidean distance from the anchor point, and (b) considering a de-biasing of the effect of positive samples (that should be near to the anchor). Experimental results show that the proposed objective outperforms the pure SimCLR and its de-biased-only variants in many but not all cases in image, graph and text domains.

Overall, I found the paper is well-written, with a clearly-motivated method. I also liked their theoretical analysis on the method. Experimental results can be a weakness of the paper for some readers in several aspects: e.g., lack of large-scale experiments, and the mixed results on comparing the method with the "debiased" baselines. Nevertheless, I think it is ok considering that the paper instead has provided an extensive evaluation over multiple modalities.

- Section 5.1: There could be a more justification on why the proposed method works not better than "Debiased" on CIFAR-10. I thought Section 6.1 would handle this, but apparently it seems not.
- Figure 4: Why the results on CIFAR-10 are not presented? It would be nice to give more information for the readers on why the method works less effectively on that dataset.
- It was a bit confusing for me to follow Eq. 3 or 4 before I noticed that q is conditioned on x. Although the paper mention that x will be omitted in the context (p2), I think there could be some place to remind this for better clarity.
- The paper could further provide a practical guide on how to choose tau^+.

---

> ### Author Response · Authors · 2020-11-19
> **We have made several improvements based on your feedback**
>
> We are grateful for the time taken to thoughtfully consider and evaluate our work, and the encouraging assessment. We would like to discuss the points you raise, and outline improvements we have made based on your review:
>
> > why [isn’t] the proposed method ... better than ‘Debiased’ on CIFAR10
>
> We speculate that the reason CIFAR10 doesn’t do as well as CIFAR100 and STL10 essentially is because it is the “easiest” problem: there is insufficient diversity in inputs to make one negative significantly harder than another. This is corroborated by Figure 10 (appendix, new version of paper) which shows that the debiasing already discriminates positive vs. negative pairs of inputs very well (better than CIFAR100 & STL10), leaving little room for hard negatives to help.
>
> > Figure 4... results on CIFAR10
>
> Indeed it is still valuable to see beta ablation for CIFAR10. The new version of the paper includes these results (Fig.7, appendix). The plot shows the same qualitative result (performance decays for large beta, best for intermediate/small beta).
>
>
> > Confusing… to follow Eq. 3 or 4
>
> We have rectified this by clearly stating that $q$ is conditioned on $x$ at the point we define $q$ (beginning of Section 3.1), and explain that we drop this dependency from the notation only for convenience.
>
>
> > A practical guide on how to choose $\tau^+$
>
> Thank you for pointing out this important question for practitioners. We have included a short discussion (beginning of Section 5) to help guide a user. We summarize this here: there are two ways to select $\tau^+$: either by estimating it from data, or by treating it is a hyperparameter. The first approach would require labeled data to be at hand prior to contrastive training.
>
> ----
> Post-deadline, we extended the theoretical analysis: we discovered that it is possible to build on top of the ball-packing theorem to shed further light  on generalization properties of contrastive embeddings learned using hard negatives. In the newly uploaded version of the paper this comes in the form of a new theorem - Theorem 5 - which shows that approximate minimizers of the objective are also well clustered (similarly to global minimizers as shown by Thm. 4) and that this leads to desirable generalization properties.

---

### Official Review · AnonReviewer3 · 2020-10-28
**Review of "Contrastive Learning with Hard Negative Samples"**

**Rating:** 6
**Confidence:** 4

**Review:**

Update:

The revisions are good. The paper is very easy to follow and most of the story is pretty clear. Theory sections are clearer as well. So I'll improve my score as the authors followed through with both mine and other reviewer's comments. There's one hitch: Pr1, the one having to do with the labels, is not well substantiated in the paper, though it gets first-class treatment in Fig 1 as well as being one of 2 main principles guiding this method. Some of this is carried over from the de-biasing work, but I have concerns that there's essentially a trade-off between hardness and label distribution depending on beta. Unfortunately, this paper does no empirical analysis on the labels in q, and I worry that readers may be mislead that something close to Fig 1 might happen in practice.

====

This paper essentially extends / iterates on [1], using a different proposal distribution for the infoNCE loss. In this case, it's a type of exponential distribution with a weighting hyperparameter beta which allows the model to concentrate the distribution of negative samples around those which have high score. As they are building off of [1], they also get the debiasing effect from that work.

This is a fine idea and it seems validated by the experiments. My first comment is there's a few missing works. 1) [2] also showed that using different proposal distributions for the infoNCE loss to increase the hardness of the contrastive task. In that case, they cover several types of distributions. 2) More or less I disagree that contrastive learning need be unsupervised. There are existing works (e.g., [3] called CMDIM) that use labels to generate a mixture distribution of positive samples that come from the same instance as well as from other instances of the same class. They also omit same class from negative samples. Really, contrastive learning should be able to leverage any information available to generate similar tasks for the model to solve. This could be thought of as a hybrid between contrastive and metric learning, but I see no reason to complicate the naming of things.

Other remarks:
* Much of the intro regards contrastive learning, augmentation, and mutual information. As such, proper credit needs to be given to [4].
* The use of the word "label" on the last paragraph second page is a little confusing as the context here is unsupervised learning. Could you clarify what you mean here? (same as first full paragraph 3rd page)
* Q is a constant brought in from [1], it would be nice to know what this means in this paper and why it's introduced without having to read that one work.
* page 4 "rejection sampling... could be slow". Are there any experiments to support this? Rejection sampling would be easy to implement, so I'm not sure I'm comfortable with discounting it so easily here.
* You are making a lot of empirical estimations of things, particularly of the partition functions. I feel like bias / variance analysis is necessary here.
* Page 5 you introduce the debiasing idea, which could be made clearer in this work.
* The explanation at the end of 4.1 is very confusing to me or not clear. Could you explain more clearly why q_beta represents a tractable approximation for large beta?
* Page 5 "representation generalizable": generalizable to what? Downstream tasks, test distribution? I'm not sure what follows with the ball packing relates to generalization.
* Figure 2: why were those betas chosen?
* 6.1: this would be a great place to mention CMDIM [2]
* For the annealing experiments, wouldn't it make more sense to anneal beta up (progressively harder) than down?

One last remark is the paper borderlines iterative from [1]: there are some interesting new things, but not nearly as much analysis or comparisons to other proposal distributions to make this a significant contribution. Including some of the baselines from [2] might help as well as more bias / variance analysis of the empirical estimators used here would help.

[1] Chuang, Ching-Yao, et al. "Debiased contrastive learning." arXiv preprint arXiv:2007.00224 (2020).
[2] Wu, Mike, et al. "On Mutual Information in Contrastive Learning for Visual Representations." arXiv preprint arXiv:2005.13149 (2020).
[3] Sylvain, Tristan, Linda Petrini, and Devon Hjelm. "Locality and compositionality in zero-shot learning." arXiv preprint arXiv:1912.12179 (2019).
[4] Bachman, Philip, R. Devon Hjelm, and William Buchwalter. "Learning representations by maximizing mutual information across views." Advances in Neural Information Processing Systems. 2019.

---

> ### Author Response · Authors · 2020-11-20
> **Thank you for your feedback. We have made several improvements based on your suggestions**
>
> We are grateful for the suggested literature - which we have added in - and points of clarity that help improve the exposition of our work. We would like to go through your list of remarks one by one to explain how we have improved the paper by incorporating each into the updated version:
>
> > The use of the word "label" … is a little confusing
>
> We have rephrased the sentence intro to state: “cannot exploit true similarity information since there is no supervision” to make it clear that it is a lack of “true” similarity information that is the issue. On page 3 we begin to use the term “label” after we have introduced the formalism we are working in. In this formalism we introduce a label space, allowing us to speak of “labels”
>
>  > It would be nice to know what [$Q$] means in this paper
>
> We added an explanation of $Q$ (see Eqn 1, where $Q$ is first introduced).
>
> > Rejection sampling … could be slow
>
> We have modified this sentence to instead motivate the importance sampling method by a desire to avoid modifying the procedure by which batches are sampled. Our method achieves the same goal as rejection sampling (i.e. estimating the expectation) but has this added element of algorithmic “simplicity” since the only part of the training pipeline that changes is the objective function.
>
> > Bias/variance analysis
>
> We think this is an interesting question. In answer to it, we provide a new lemma (Lemma 11, appendix) which shows that the variance of the random variable $X=e^{f(x)^\top f(v)}$ where $x \sim p$ and $v \sim q_\beta^+$ is bounded by $O(\mathcal L_\text{align}(f))$ (where $\mathcal L_\text{align}$ is the alignment loss from Wang & Isola ICML 2020). Since Wang & Isola show that the contrastive objective optimizes $\mathcal L_\text{align}(f) + \mathcal L_\text{unif}(f)$ we expect $\mathcal L_\text{align}(f)$ to be small, meaning the random variable X has low variance. Meanwhile, the empirical estimate over $N$ negative samples is fairly well concentrated since the random variable is bounded and $N$ is fairly large (e.g. $N=510$).
>
>
> > The debiasing idea ... could be made clearer in this work
>
> We have reformulated our introduction of this work. We first introduce the idea of debiasing in the related works section (pg. 2), stating that debiasing “corrects for the fact that not all negative pairs may be true negatives.  It does so by taking the viewpoint of Positive-Unlabeled learning, and exploits a decomposition of $p^-$.” We also re-introduce the intuitive idea “[debiasing] amounts to correcting for the fact that some samples in a negative batch sampled from p will have the same label as the anchor.”
>
> > The explanation at the end of 4.1 is very confusing … could you explain more clearly why $q_\beta$ represents a tractable approximation for large $\beta$?
>
> We have reworked this paragraph to avoid the previous ambiguities and only use precise (and verified) mathematical claims. By “tractable approximation” we were referring to two things: first, that the resulting objective with $q_\beta$ plugged in is easily approximately evaluated (using importance sampling), and second we were referring to Prop 3., which states that as beta goes to infinity the objective value converges to the value under the worst case negatives distribution.
>
>  > ‘Representation generalizable’... I'm not sure what follows with the ball packing relates to generalization.
>
> Intuitively, since we expect the classes to be well clustered, it should be possible to define a classifier on the hypersphere that attains low generalization error. We worked post-deadline to obtain a new theorem - Theorem 5 in the newly updated version of the submission - which makes formal the connection between minimizing the objective up to some small epsilon, and being “well clustered”. The new analysis shows that a simple 1-nearest neighbor classifier attains low generalization error. Since low 1-nearest neighbor risk implies being well clustered, this also suggests why linear classifiers (i.e. separating hyperplanes through the sphere) also generalize well.
>
> >Figure 2... why those betas?
>
> We have added a sentence stating that beta essentially has to be tuned as a hyperparameter, and pointing to Figure 5 for some intuition in how to do this: namely, picking a “relatively small” but strictly positive beta should do fine.
>
> >6.1 … would be a great place to mention [2]
>
> We have added this in.
>
> >For the annealing experiments, wouldn't it make more sense to anneal beta up?
>
> We agree that annealing “up” also makes sense. We ran preliminary experiments and, interestingly, we found the results to be quite comparable to those reported for annealing “down” - e.g. readout acc of 86.76% annealing up to $\beta=6$, and acc 87.05% annealing up to $\beta=2$. For now, we have included a sentence (in appendix) simply stating that annealing “up” achieves similar results; but will continue to run the full batch of tests and plan to include them in the final version.

---

### Official Review · AnonReviewer2 · 2020-10-29
**Contrastive Learning with Hard Negative Samples**

**Rating:** 6
**Confidence:** 4

**Review:**

In this paper, the authors mainly study how to sample good/informative negative examples for contrastive learning. The key challenge is the unsupervision in contrastive methods.  They propose a new unsupervised method to select the hard-negative samples with user control. The experimental results on three modalities (images, graph and sentences) show the effectiveness of their method.

Strength.
This paper is well written and easy to follow.
This work mainly focuses on how to sample informative negative examples in contrastive learning. Targeting at this problem, they propose a new hard negative sampling with theoretical analysis.
The authors also do lots of comparisons on multiple modalities dataset to indicate the effectiveness of their method. Besides, they also do interesting ablation studies, in Section 6, for example, to investigate the effect of the more harder samples and debiasing.

Weakness.
This is an interesting and important problem. I have noticed that, there is another paper released on ArXiv to deal with this similar problem at the same time. (Hard Negative Mixing for Contrastive Learning). It is unfair for the authors to do experimental comparison, while I might suggest that if possible the authors might just do an analysis comparison without any experimental results.

Visualization. If possible, the authors might give more visualization examples, such as t-SNE or some numerical values to show the distances between anchor and the sampled hard samples with different hardness on the batch on three modalities, including images, graph and sentences. For example, just like the Figure 1, the authors might show some sampled negative sentences by typical method and their method. For the proposed hardness, maybe they could show some sampled sentences with different hardness, as an additional visualization. Besides, if possible, they could also show some similar figures like Figure 5.

---

> ### Author Response · Authors · 2020-11-20
> **We have added more visualizations**
>
> We are grateful for your considered evaluation, and appreciate the constructive ideas for improvement. We have added material based on your useful feedback and suggestions. The main action points are:
>
> > If possible, the authors might give more visualizations
>
> We have added a new figure (Figure 12, appendix), which is a “real data” version of Figure 1. The figure shows that hard negatives are semantically much more similar to the anchor than uniformly sampled negatives. Indeed, hard negatives possess many similar characteristics to the anchor, including texture, colors, and containing animals (as opposed to machinery). The figure does not account for the debiasing that is added.
>
> > could also show some similar figures like Figure 5. (histograms)
>
> We have added figures (Figures 9, 10, appendix) showing the same result as Figure 5 but on the CIFAR100 and CIFAR10 datasets. They tell a similar story as Figure 5 does for STL10.
>
> > there is another paper released on arXiv to deal with this similar problem
>
> We have modified the related works section to mention this related work, which was new to us and we found to be very interesting and insightful work. We have also added a small discussion comparing our methods at the end of the appendix  (due to space restrictions). The main points of difference are: 1) MoCHi considers the benefit of hard negatives, but does not consider the possibility of false negatives, 2) MoCHi introduces three parameters (N, s, s’), while we introduce two (beta, tau_plus). If we drop the “avoiding false negatives” aspect of our work ,then we only need to tune one parameter: beta, 3) our method introduces 0 computational overhead, whereas MoCHi involves a small amount of extra computation.
>
>
> ----
>
> We also found it interesting to also study the optimization speed of hard sampling vs SimCLR. Figure 11 shows these results, which show that hard sampling achieves a given readout accuracy after much fewer epochs than SimCLR. This suggests another use of hard sampling is to reduce the number of epochs (training time) for an embedding to reach a certain readout accuracy.

---

### Author Response · Authors · 2020-11-24
**A summary of updates to the paper**

First, once again we would like to thank all reviewers for their efforts and considered thoughts in reviewing our paper. We are grateful for the encouraging feedback, and for the suggestions  of ways to improve the paper. The purpose of this message is to give a single post containing a synopsis of the updates that we have made to the paper based on your feedback.

- A new theoretical result studying the generalization of contrastive embeddings under worst-case negatives (Theorem 5). The result, building on the ball-packing result, establishes a generalization guarantee by showing classes are well clustered.

- Bias/variance analysis for the expectation approximated using $M$ positive samples (where $M=1$ in experiments). We show that the variance of the random variable $e^{f(x)^\top f(v)}$ with $x \sim p$ and $v \sim q_\beta^+$ is bounded by $\mathcal L_\text{align}(f)$ (where $\mathcal L_\text{align}$ is the alignment loss introduced in appendix, in line with Wang & Isola 2020). This observation suggests the variance will reduce over time as  $\mathcal L_\text{align}(f)$ decreases, leading to better empirical estimates.

- Added missing annealing results to Fig. 4 for STL10.

- MoCo with hard negatives: Fig. 6 in appendix shows that hard negative sampling can improve downstream generalization even when using vey large negative batch sizes.

- Added histograms of cosine similarity of similar/dissimilar pairs for CIFAR10 and CIFAR100 (Figs. 9 and 10, analogous to Fig 5. in original submission). They both tell a similar story to Fig. 5.

- Added plot of trajectory of generalization as training evolves. Results show that on STL10 hard negative sampling takes only 60 epochs to reach the same linear readout performance as SimCLR does in 400 epochs.

- Added Fig. 12, a "real data" version of Figure 1. The figure shows that hard negatives are semantically similar to the anchor.

- Added related work identified by reviewers, and fixed unclear segments: e.g. discussion of interpolation result (Prop. 3), and the introduction of $q_\beta$.

---

### Decision · Program_Chairs · 2021-01-07
**Final Decision**

**Decision:**

Accept (Poster)

**Comment:**

This paper proposes a contrastive learning framework that leverages hard negative samples for self-supervised training. The proposed framework is theoretically analyzed and its efficacy is examined on several datasets/problems. A group of expert reviewers reviewed the paper and provided positive ratings for this paper. I agree with the reviewers and I recommend accepting this submission.

One of the main discussion points among the reviewers was to what degree Pr1 is "approximately" satisfied in the proposed framework. There are several approximations in this paper that are not fully analyzed.  Some of these approximations could be examined assuming that labeled data is available during training. For example, $p_x^+$ is approximated using a set of semantics-preserving transformations. In practice, the distribution induced by augmenting $x$ is very different than the distribution that samples from the instances in the class of $x$. The effect of this approximation could be easily examined by sampling from true class labels. Additionally, it would be very helpful to visualize how $q$ samples from the negative instances and how much it follows Pr1.

I would like to ask the authors to add a small limitations section to the final camera-ready version that lists all the assumptions and approximations made in this paper. Please provide a high-level analysis on how such assumptions could be validated or such approximations could be measured if labeled data or additional information was provided. This discussion is extremely important for future practitioners to understand the basic assumptions that may not hold in reality and it will enable them to improve upon this work.

---

> ### Comment · ~Joshua_David_Robinson1 · 2021-03-17
> **Following up on the additional discussion**
>
> Thank you for the feedback and suggestions for the camera ready version.
>
> - We have added an additional discussion of the approximations we make. We discuss each point as they arise in the paper, but have also added a single, centralized paragraph where all these points can be found in one place.
> - On the point of running experiments using true positives to examine the effect, we refer the reader to Fig. 4 (middle). We find that using true positive yields linear readout performance that increases monotonically with concentration parameter $\beta$. This point is important for practitioners, and we include a discussion.
> - Finally, on the point of visualizing samples from $q$: we refer the reader to Fig. 12 (appendix) for a comparison of real samples generated according to $q$ compared to uniform samples.